# Breaking the Self-Confirming Loop: Diagnosing and Mitigating Systemic Reward Bias in Self-Rewarding RL

Chuyi Tan [1] [*]   Peiwen Yuan [1] [*]   Xinglin Wang [1]   Yiwei Li [1]   Shaoxiong Feng [2]   Yueqi Zhang [1]   Jiayi Shi [1]
Ji Zhang [1]   Boyuan Pan [2]   Yao Hu [2]   Kan Li [1]

## Abstract

Reinforcement learning with verifiable rewards (RLVR) efficiently scales the reasoning ability of large language models (LLMs) but is bottlenecked by scarce labeled data. Reinforcement learning with intrinsic rewards (RLIR) offers a scalable alternative via self-rewarding, yet often suffers from instability and inferior performance. We trace this gap to a systemic bias in confidence-coupled self-rewarding: the model tends to over-reward high-confidence mistakes, forming a **self-confirming loop**. We quantify this feedback-loop bias with three metrics: reward noise magnitude ($\rho_{\text{noise}}$), policy–reward coupling ($\rho_{\text{selfbias}}$), and over-/under-reward skew ($\rho_{\text{symbias}}$). Our analyses show a compounding effect where strong coupling amplifies confidence-conditioned errors and drives a drift toward over-reward, leading to instability and a lower performance ceiling. To mitigate this, we propose reinforcement learning with ensembled rewards (**RLER**), which aggregates diverse models with adaptive reward interpolation and disagreement-aware rollout selection to reduce coupling and suppress over-reward drift. Extensive experiments show that RLER improves by 6.2% over the best RLIR baseline and is within 3.6% of RLVR, while exhibiting stable scaling on unlabeled samples.

## 1 Introduction

Reinforcement learning with verifiable rewards (RLVR) can efficiently scale the reasoning capabilities of large language models (LLMs) (Guo et al., 2025; El-Kishky et al., 2025;

Team et al., 2025; Gao et al., 2023). However, it is bottlenecked by the scarcity of labeled data, limiting continued data scaling (Gunjal et al., 2025; Zhang et al., 2025c). In contrast, reinforcement learning with intrinsic rewards (RLIR, also known as self-rewarding RL), in which the policy model assigns reward signals to itself, enables sustainable scaling in unlabeled settings (Huang et al., 2025; Zuo et al., 2025). It not only reduces annotation cost but is also particularly valuable in domains with abundant unlabeled data yet scarce supervision, such as private corpora or industrial applications.

Nevertheless, its performance gain and stability still fall short of RLVR (Shafayat et al., 2025; Zhang et al., 2025c). We trace this gap to **a systemic reward bias** in self-reward estimation. Under RLIR, reward estimation is strongly coupled with the policy's confidence, yielding an asymmetric error pattern: reward errors stay small for confident correct rollouts but become large for confident mistakes. This asymmetry forms a self-confirming loop in existing RLIR methods(Zuo et al., 2025; Huang et al., 2025), where biased rewards accumulate over training and drift toward over-rewarding, leading to unstable optimization and a lower performance ceiling.

We introduce **three metrics that characterize the mechanics of this feedback loop**: (i) reward noise rate $\rho_{\text{noise}}$: measures the absolute magnitude of reward-estimation bias; (ii) self-feedback bias rate $\rho_{\text{selfbias}}$: measures how tightly reward estimates are coupled to the policy, i.e., how strongly bias is reinforced by the loop; and (iii) symmetry bias rate $\rho_{\text{symbias}}$: measures whether the bias is skewed toward over-reward or under-reward, i.e., the drift direction. Based on these metrics, our controlled analyses reveal **two key drivers of RLIR's failure modes**. First, excessive reward noise ($\rho_{\text{noise}}$) slows convergence and can even collapse training. Second, policy–reward coupling ($\rho_{\text{selfbias}}$) reinforces confidence-conditioned errors and destabilizes reward estimation across instances. $\rho_{\text{symbias}}$ further indicates that this instability typically drifts toward over-reward, which is more damaging than under-reward in our analyses.

Therefore, to sustain stable scaling, the reward-estimation space should simultaneously satisfy: (i) Accuracy: keep

---

[*]Equal contribution  [1]School of Computer Science, Beijing Institute of Technology, Beijing, China  [2]Xiaohongshu Inc, China. Correspondence to: Boyuan Pan <panboyuan@xiaohongshu.com>, Kan Li <likan@bit.edu.cn>.

*Proceedings of the $43^{rd}$ International Conference on Machine Learning*, Seoul, South Korea. PMLR 306, 2026. Copyright 2026 by the author(s).

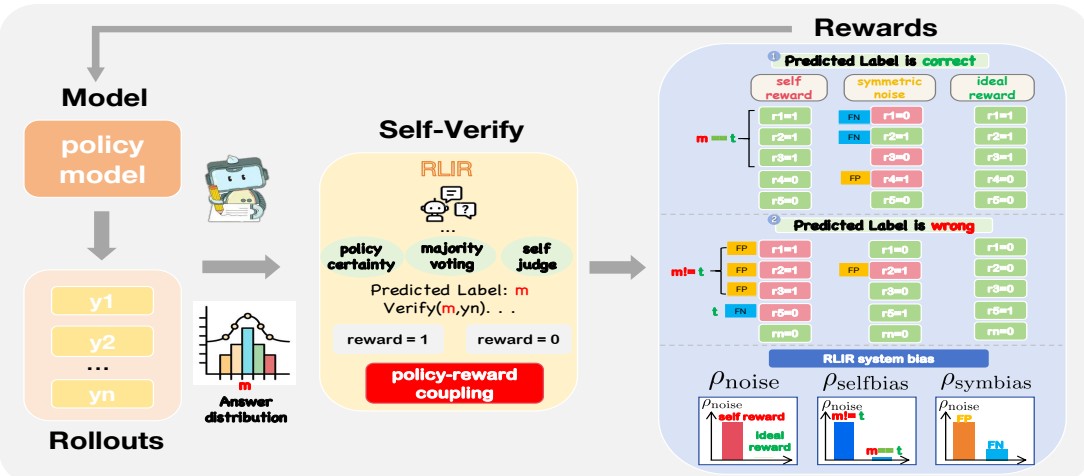

*Figure 1.* Overview of RLIR and the self-confirming reward loop. A policy samples multiple rollouts, derives intrinsic rewards from its own outputs, and updates on these rewards; when reward estimates are confidence-coupled, high-confidence mistakes can be reinforced over training.

low $\rho_{\text{noise}}$ (avoiding collapse under high noise). (ii) Unbiasedness: prevent asymmetric drift toward over-reward ($\rho_{\text{symbias}}$). (iii) Robustness: decouple reward estimates from policy confidence ($\rho_{\text{selfbias}}$) to prevent self-confirmation.

To mitigate this systemic bias, we propose reinforcement learning with ensembled rewards (**RLER**). RLER breaks the confirming loop of single-policy self-rewarding by estimating rewards with an ensemble of diverse policies, and is designed to satisfy the three desiderata above: it reduces reward noise (Accuracy), attenuates confidence-coupled self-reinforcement (Robustness), and counteracts drift toward over-reward (Unbiasedness). Concretely, RLER instantiates these goals through three mechanisms: (i) **Ensemble-based Unified Rewarding**, which aggregates rewards across diverse policies to reduce reliance on any single policy's confidence; (ii) **Adaptive Soft-reward Interpolation**, which adaptively blends hard and soft rewards to stabilize learning under varying confidence; and (iii) **Disagreement-Aware Rollout Selection**, which uses ensemble disagreement to surface and penalize high-confidence mistakes, thereby correcting the over-reward skew. Finally, we merge the ensemble into a single deployable policy, incurring no additional inference cost at deployment.

To systematically evaluate RLER, we conduct extensive experiments across diverse tasks, datasets and models. The results show that RLER improves by +6.2% over the best RLIR baseline, and is only 3.6% below the RLVR setting. Moreover, RLER effectively mitigates the systemic reward bias, significantly reduces $\rho_{\text{noise}}$, $\rho_{\text{selfbias}}$, and $\rho_{\text{symbias}}$. It also exhibits stable scaling with unlabeled data. After model merging, the final deployable policy achieves higher accuracy and stability with no additional inference cost.

## 2 Related Works

**Reinforcement learning with intrinsic rewards (RLIR)** RLIR reduces reliance on human labels by generating policy rollouts and deriving rewards from intrinsic signals. Existing methods can be broadly grouped by the *source of the reward signal*: (i) *Agreement-based* methods leverage self-consistency by taking rollouts consensus (e.g., majority vote) as a pseudo label, which is then verified to produce reward signals for training (Zuo et al., 2025; Huang et al., 2025; Zhang et al., 2025c); (ii) *Confidence/uncertainty-based* methods derive intrinsic reward signals from the policy's own confidence/uncertainty statistics, using these signals as scalar rewards without requiring external labels (Zhang et al., 2025a; Agarwal et al., 2025; Li et al., 2025; Zhao et al., 2025); and (iii) *LLM-as-a-judge* methods obtain reward signals from a judging process (e.g., self-judge or self-play) to improve coverage and verifiability (Arnesen et al., 2024; Yuan et al., 2024; Xiong et al., 2025). The first two families derive rewards from a single policy's own outputs or its lagged reference versions, often coupling rewards to the policy and amplifying confidence-conditioned errors. They also typically adopt fixed reward designs (e.g., hard vs. soft), which can affect stability. RLER instead estimates rewards with an ensemble, using adaptive interpolation and disagreement-aware selection to reduce coupling and drift.

**Learning with Noisy Labels** Learning with noisy labels aims to improve robustness under corrupted supervision (Frénay & Verleysen, 2013; Zhang et al., 2016a; Nigam et al., 2020). Classic formulations typically categorize noise as instance-independent (often symmetric or asymmetric) or instance-dependent (Song et al., 2022; Zhang et al., 2016b). Recent analyses show that self-generated supervision can be unstable and may collapse when the feedback signal is not externally grounded (Zhang et al., 2025b). We further find

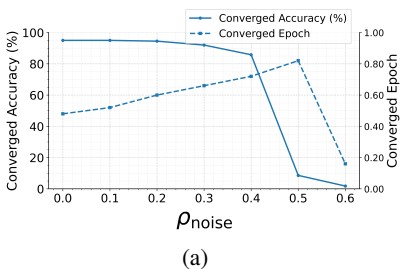 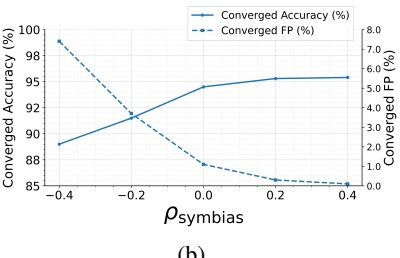 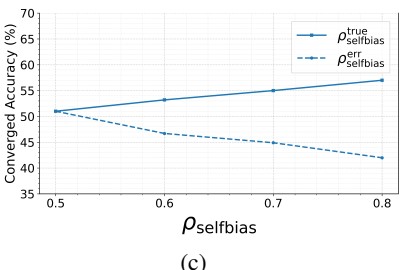

(a)         (b)         (c)

*Figure 2.* Controlled decoupling study on the arithmetic dataset. We independently vary reward noise magnitude $\rho_{\text{noise}}$, over-/under-reward skew $\rho_{\text{symbias}}$, and policy–reward coupling $\rho_{\text{selfbias}}$ to isolate how each factor affects RLIR training dynamics.

that, In RLIR, reward noise is induced by the policy itself and is tightly coupled to the policy's outputs/confidence; it can be non-stationary over training and exhibit a directional skew between over- and under-reward. These properties make it different from standard label-noise settings and motivate explicit diagnostics of noise magnitude, coupling, and skew.

## 3 Preliminary

In this section, we first specify the RLIR training loop and the reward-estimation setting. We then define three metrics: $\rho_{\text{noise}}$, $\rho_{\text{selfbias}}$, and $\rho_{\text{symbias}}$ to quantify reward-estimation error, policy–reward coupling, and over-/under-reward skew, respectively. Finally, we conduct a controlled decoupling experiment that isolates these factors to study how each one affects RLIR training dynamics.

### 3.1 RLIR Training Loop and Reward Estimation

**RLIR training loop.** RLIR iterates over three steps: (i) sample rollouts from the current policy $\pi_\theta$ given a query $x$; (ii) estimate intrinsic rewards for the rollouts using a self-reward estimator $\mathcal{R}$; and (iii) update the policy using a policy-gradient objective (e.g., GRPO (Shao et al., 2024)).

**Reward Estimation in RLIR.** In this work, we instantiate RLIR in a GRPO-style *group setting* with group size $G$, sampling $\mathcal{Y}_\theta(x) = \{y_i\}_{i=1}^G$ and assigning per-rollout rewards $\{\tilde{r}_i\}_{i=1}^G = \mathcal{R}(\mathcal{Y}_\theta(x))$. In what follows, we adopt two representative *agreement-based* estimators: a *soft* rule and a *hard* rule, which will be used throughout the paper. Let $\ell : \mathcal{Y} \rightarrow \{0, \dots, L-1\}$ be a labeling map and define the empirical answer distribution

$$p_j \;=\; \frac{1}{G} \sum_{i=1}^G \mathbf{1}[\ell(y_i) = j]. \tag{1}$$

We consider a soft estimator, Frequency-based (Freq), which assigns each rollout the empirical probability of its label, and a hard estimator, Self-Consistency (SC), which binarizes

the majority label $m = \arg\max_j p_j$:

$$\mathcal{R}_{\text{Freq}}(\mathcal{Y}_\theta(x)) = \big\{\, p_{\ell(y_i)} \,\big\}_{i=1}^G, \tag{2}$$

$$\mathcal{R}_{\text{SC}}(\mathcal{Y}_\theta(x)) = \big\{\, \mathbf{1}[\ell(y_i) = m] \,\big\}_{i=1}^G. \tag{3}$$

The resulting rewards are then converted to advantages and used to update the policy.

### 3.2 Reward noise rate

Let $t$ be the ground-truth label for query $x$. For each rollout $y_i$, we define the oracle reward as $r_i^\star = \text{verify}(\ell(y_i), t) \in \{0, 1\}$, where $\text{verify}(\cdot)$ returns 1 iff the rollout answer $\ell(y_i)$ matches $t$. We use $r_i \in [0, 1]$ to denote the actual reward used for policy updates. We measure the reward noise rate as the mean absolute deviation from the oracle reward:

$$\rho_{\text{noise}}(x) \;=\; \frac{1}{G} \sum_{i=1}^G \big| r_i - r_i^\star \big|. \tag{4}$$

### 3.3 Self-feedback bias rate

RLIR induces *policy–reward coupling*: the policy's answer distribution shapes the reward distribution. Here, $r_i$ denotes the final reward used for updating the policy, while the policy-based reward $\tilde{r}_i$ denotes the score that the rollout would receive if evaluated only by its own source policy using that policy's local rollout distribution and intrinsic-reward rule. We quantify this coupling by the *self-feedback bias rate*:

$$\rho_{\text{selfbias}}(x) = 1 - \frac{1}{G} \sum_{i=1}^G |r_i - \tilde{r}_i| \tag{5}$$

**Correctness–confidence effect.** Let $p_t$ and $p_m$ be the empirical probabilities of the ground-truth and majority label under query $x$. (i) $m = t$ (Alignment): High confidence reduces noise. SC achieves $\rho_{\text{noise}} = 0$, while Freq is bounded by $1 - p_m$, vanishing as $p_m \rightarrow 1$. (ii) $m \neq t$ (Misalignment): High confidence amplifies error. SC yields $\rho_{\text{noise}} = p_m + p_t$, implying that stronger consensus on mistakes worsens bias. Soft rewards mitigate this by distributing credit; we prove that Freq yields lower reward error than SC when the ground-truth label has the second-largest mass, i.e., $p_t \geq \max_{j \notin \{m, t\}} p_j$ (full proof in Appendix C).

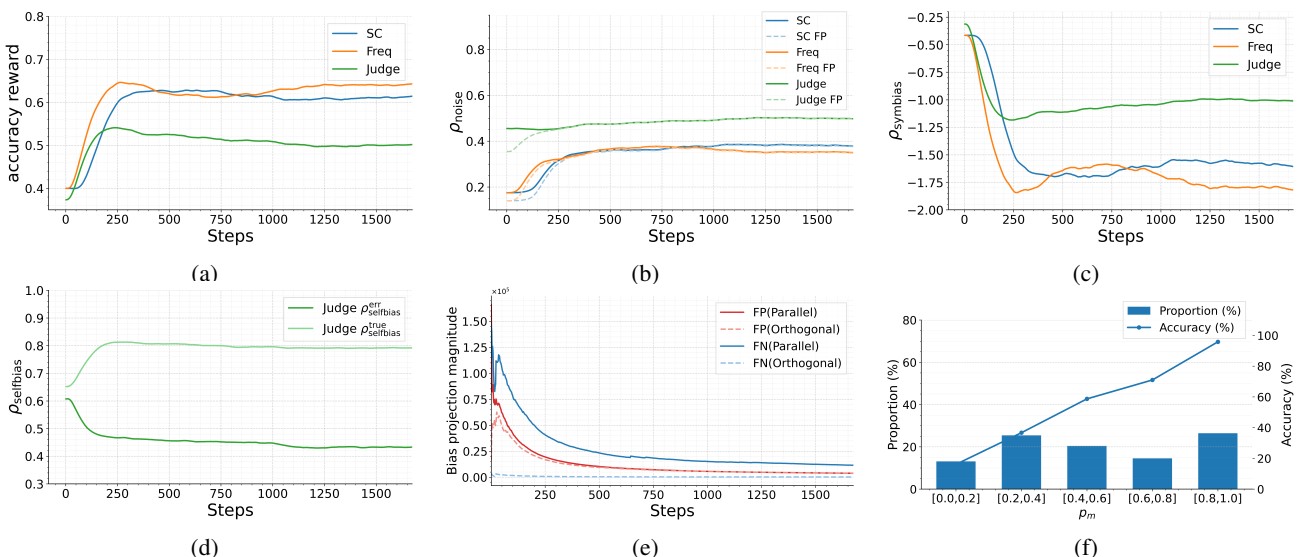

*Figure 3.* Bias dynamics of representative RLIR methods on the arithmetic dataset. Single-policy intrinsic rewards accumulate reward noise, drift toward FP-dominated over-reward, and remain strongly coupled with the policy's own confidence, explaining their unstable training behavior.

### 3.4 Symmetry-bias rate

Compared to symmetric noise, RLIR's policy–reward coupling introduces a directional bias between over-reward and under-reward. We term the directional components false-negative (FN): under-reward relative to the oracle; and false-positive (FP): over-reward. With $(u)_+ = \max\{u, 0\}$,

$$\mathrm{FN}(x) = \frac{1}{G}\sum_{i=1}^{G}(r_i^\star - r_i)_+, \quad \mathrm{FP}(x) = \frac{1}{G}\sum_{i=1}^{G}(r_i - r_i^\star)_+. \tag{6}$$

We define the *Balance Ratio (BR)* as $\mathrm{FN}/\mathrm{FP}$. Under an ideal symmetric noise assumption (where reward noise is independent of rollout correctness), the BR represents the class-imbalance ratio: $\mathrm{BR}_{\mathrm{sym}}(x) = p_t/(1 - p_t)$, where $p_t$ is the oracle accuracy. We measure the *symmetry bias rate* as the deviation from this symmetric baseline:

$$\rho_{\mathrm{symbias}}(x) = \mathrm{BR}_{\mathrm{IR}}(x) - \mathrm{BR}_{\mathrm{sym}}(x). \tag{7}$$

A negative $\rho_{\mathrm{symbias}}$ indicates a drift toward over-reward (FP dominant), while positive indicates under-reward.

### 3.5 Decoupling experiment

We conduct a systematic set of experiments to separately analyze the effects of three metrics on RLIR training and to identify the causes of biased and unstable reward estimation.

**Experiment Setup.** We construct a controlled testbed using a synthetic arithmetic dataset (375k samples, see Appendix B.1.1) with QWEN2.5-1.5B-INSTRUCT as the base policy. To isolate the metrics, we synthesize reward signals from the oracle $\{r_i^\star\}$ via a three-stage injection process: (i)

injecting symmetric flips to control magnitude ($\rho_{\mathrm{noise}}$); (ii) applying asymmetric flipping to modulate the FN/FP balance ($\rho_{\mathrm{symbias}}$); and (iii) coupling rewards with the policy's real-time predictions to adjust feedback strength ($\rho_{\mathrm{selfbias}}$). We train models under these synthesized reward landscapes and observe the following key dynamics:

**Findings 1: $\rho_{\mathrm{noise}}$ governs the convergence performance and speed.** As $\rho_{\mathrm{noise}}$ rises, the performance ceiling drops and training shifts from stable convergence to collapse; within the transition regime, higher noise monotonically slows convergence.

**Findings 2: Over-reward is more detrimental than under-reward.** With $\rho_{\mathrm{noise}}$ held constant, as $\rho_{\mathrm{symbias}}$ increases, the imbalance shifts from an over-reward bias to an under-reward bias; meanwhile, the converged performance rises, indicating that over-rewarding is more detrimental. Further analysis shows that under-reward weakens the gradient along the correct direction, whereas over-reward assigns positive advantages to incorrect outputs; both effects dampen correct updates and introduce a near-orthogonal gradient bias (as seen in Fig. 2(b) and Fig. 3(e)).

**Findings 3: High $\rho_{\mathrm{selfbias}}$ amplifies both correct and incorrect updates.** As seen in Figure 2(c), where we shorthand $\rho_{\mathrm{selfbias}}^{\mathrm{true}} := \mathbb{E}[\rho_{\mathrm{selfbias}}(x) \mid m = t]$ and $\rho_{\mathrm{selfbias}}^{\mathrm{err}} := \mathbb{E}[\rho_{\mathrm{selfbias}}(x) \mid m \neq t]$. when $m = t$, a higher $\rho_{\mathrm{selfbias}}^{\mathrm{true}}$ strengthens correct updates, leading to improved convergence performance; when $m \neq t$, $\rho_{\mathrm{selfbias}}^{\mathrm{err}}$ amplifies wrong-direction updates.

As seen in Figure 3, under RLIR methods, we observe similar failure patterns: reward noise accumulates over training (Fig. 3(b)), the deviation drifts toward over-reward (Fig. 3(c)), and the performance ceiling is locked (Fig. 3(a)). Moreover, SC and Frequency exhibit maximal coupling (high $\rho_{\text{selfbias}}$), while judge-based rewards reduce overall coupling but can still retain high coupling on incorrect rollouts (Fig. 3(d)).

**Findings 4: High $\rho_{\text{selfbias}}$ induces unstable reward estimation.** Prediction (majority label) correctness and confidence ($p_m$) exhibit large cross-instance variance (seen in Fig. 3(f)). We observe that RLIR methods exhibit very high policy–reward coupling, the variance propagates through this coupling, yielding unstable reward estimation.

**What reward space do we need?** Based on these insights, a reward-estimation space must simultaneously satisfy the three desiderata: (i) **Accuracy**: keeping $\rho_{\text{noise}}$ strictly below the collapse threshold. (ii) **Unbiasedness**: eliminating the asymmetric drift toward over-reward ($\rho_{\text{symbias}}$). (iii) **Robustness**: decoupling reward estimates from policy confidence ($\rho_{\text{selfbias}}$) to prevent self-confirmation loops.

## 4 RLER

Guided by the diagnostics in §3, we propose *reinforcement learning with ensembled rewards* (**RLER**) to break the self-confirming loop in single-policy RLIR. RLER constructs a *unified* reward-estimation space using a population of policies, targeting the three desiderata: Accuracy, Unbiasedness, and Robustness.

### 4.1 Ensemble-based Unified Rewarding

We replace single-policy self-rewarding with an ensemble to obtain a unified reward space that is less tied to any individual policy.

**Aggregation.** Given $K$ source policies $\{\pi_{\theta_k}\}_{k=1}^{K}$, we sample a group of rollouts $\mathcal{Y}_k(x) = \{y_{k,i}\}_{i=1}^{G}$ from each policy. Let $\ell(\cdot)$ map a rollout to its answer and define each policy's empirical answer distribution

$$p_j^{(k)}(x) \;=\; \frac{1}{G} \sum_{i=1}^{G} \mathbf{1}[\ell(y_{k,i}) = j]. \tag{8}$$

We then form the ensemble mixture

$$\bar{p}_j(x) = \frac{1}{K} \sum_{k=1}^{K} p_j^{(k)}(x),$$
$$m^{\text{EC}}(x) = \arg\max_j \bar{p}_j(x). \tag{9}$$

and pool all rollouts as $\mathcal{Y}(x) = \bigcup_{k=1}^{K} \mathcal{Y}_k(x)$.

**Why ensemble first.** Ensembling addresses the three failure modes diagnosed in §3. **First, for accuracy,** aggregating across diverse policies reduces policy-specific errors, lowering expected reward noise ($\rho_{\text{noise}}$). **Second, for robustness,** using $\bar{p}(\cdot)$ as a shared reference weakens the dependence on any single policy's confidence, reducing policy-reward coupling ($\rho_{\text{selfbias}}$). **Finally, for unbiasedness,** the mixture spreads probability mass across labels during disagreement rather than committing to confident mistakes, mitigating over-reward drift ($\rho_{\text{symbias}}$).

### 4.2 Adaptive Soft-reward Interpolation

To navigate the trade-off between the high variance of hard rewards and the low-confidence bias of soft rewards, we propose an adaptive interpolation strategy. This mechanism dynamically adjusts the estimated reward based on **unified ensemble confidence**, seeking *the optimal balance between accuracy and robustness*.

**Interpolation.** We construct the final reward $r_i$ for rollout $y_i$ by interpolating between the hard ensemble decision $r_i^{\text{H}} = \mathbf{1}[\ell(y_i) = m^{\text{EC}}]$ and the soft unified probability $r_i^{\text{S}} = \bar{p}_{\ell(y_i)}$:

$$r_i^{(\alpha)} = (1 - \alpha)\, r_i^{\text{S}} + \alpha\, r_i^{\text{H}}, \qquad \alpha \in [0, 1]. \tag{10}$$

where $\alpha(x) \in [0, 1]$ is a gate modulated by the ensemble's unified confidence.

**Unified confidence for adaptive weighting.** A raw probability average (Eq. 9) treats all policies and queries equally, failing to capture the varying query-specific confidence reflected in different policies' answer distributions and the fine-grained information at the token level. To estimate unified confidence precisely, we integrate *token-level confidence* into the ensemble.

For each source $k$, let $\mathcal{Y}_{k,j}(x)$ denote the subset of rollouts that yield answer $j$:

$$\mathcal{Y}_{k,j}(x) \;=\; \big\{\, y_{k,i} \in \mathcal{Y}_k(x) \mid \ell(y_{k,i}) = j \,\big\}. \tag{11}$$

Let $\text{conf}(y)$ be the average token probability of a rollout $y$. We define the *average answer confidence* $\bar{c}_k(j)$ for label $j$ within source $k$ as:

$$\bar{c}_k(j) \;=\; \frac{1}{|\mathcal{Y}_{k,j}(x)|} \sum_{y \in \mathcal{Y}_{k,j}(x)} \text{conf}(y). \tag{12}$$

To ensure robustness against varying difficulty across batches, we apply *Batch-wise Min-Max Normalization*:

$$\hat{c}_k(j) \;=\; \frac{\bar{c}_k(j) - \min_{\beta^-} \bar{c}_k}{\max_{\beta^+} \bar{c}_k - \min_{\beta^-} \bar{c}_k}, \tag{13}$$

where $\min_{\beta^-} \bar{c}_k$ and $\max_{\beta^+} \bar{c}_k$ denote the $\beta^-$- and $\beta^+$ quantiles of $\{\bar{c}_k(j)\}_{j \in \mathcal{J}_k}$, respectively.

We then compute a calibrated mass $s_k(j)$ by re-weighting the empirical frequency $p_j^{(k)}$ (defined in Eq. 8) with this

relative confidence:

$$S_k(j) \;=\; p_j^{(k)} \cdot c_k(j), \qquad s_k(j) \;=\; \frac{S_k(j)}{\sum_u S_k(u)}. \qquad (14)$$

Finally, we aggregate across sources to obtain a accurate and robust answer-confidence **unified ensemble estimation** and **unified ensemble confidence**:

$$\tilde{p}_j(x) \;=\; \frac{1}{K} \sum_{k=1}^{K} s_k(j), \qquad \alpha(x) \;=\; \mathrm{clip}\Big(\tilde{p}_{m^{\mathrm{EC}}}(x),\, 0,\, 1\Big).$$

### 4.3 Disagreement-Aware Rollout Selection

To further *improve accuracy and unbiasedness*, we select updates from the pooled rollouts to reduce reward noise and to counteract confidence-conditioned over-reward drift.

**Rollout allocation strategy** We treat all ensemble rollouts as one data pool and allocate updates to the $K$ sources in two ways:

- **Data sharding.** Partition the query set as $\mathcal{Q} = \bigcup_{k=1}^{K} \mathcal{Q}_k$. Model $k$ updates on queries $x \in \mathcal{Q}_k$ using the pooled rollouts $\mathcal{Y}(x) = \bigcup_{j=1}^{K} \mathcal{Y}_j(x)$.

- **Model sharding.** For each query $x$, split the pooled rollouts $\mathcal{Y}(x)$ evenly across models for updates.

Experiments show that data sharding provides stronger diversity, we therefore use it by default.

**Rollout selection strategy** Partition answer distribution into the head $m^{\mathrm{EC}}$ and the tail $\mathcal{L} \setminus \{m^{\mathrm{EC}}\}$.

$$w_{m^{\mathrm{EC}}}(x) = \alpha(x),$$
$$w_j(x) = 1 - \tilde{p}_j(x), \qquad j \neq m^{\mathrm{EC}}.$$

Let $b(x)$ be the resulting per-query update budget after reweighting:

$$\mathrm{take}_y \;=\; \min\Big\{ G,\, \mathrm{round}\big(G \cdot w_y(x)\big)\Big\}, \qquad b(x) = \sum_y \mathrm{take}_y$$

This design is tightly coupled with the interpolated reward in Eq. 10. For the majority label $m^{\mathrm{EC}}$, rollouts receive the hard reward $r^{\mathrm{H}}$, hence we scale the head budget with $\alpha(x)$ to emphasize updates only when the ensemble consensus is reliable. For tail labels ($j \neq m^{\mathrm{EC}}$), $r^{\mathrm{H}} = 0$ and updates rely on the soft term $(1-\alpha)\, r^{\mathrm{S}}$; when consensus is reliable, larger $\alpha(x)$ naturally suppresses these tail updates, while lower confidence preserves more soft credit for plausible minority answers. We therefore avoid overly suppressing tail labels that may correspond to the correct answer under disagreement, while low-frequency noise is naturally attenuated by its small $r^{\mathrm{S}}$ together with the per-policy rollout-budget cap.

### 4.4 Ensemble-to-Single Consolidation

To enable single-model deployment with no additional inference overhead, we consolidate the $K$ trained policies into one model via Ties-Merging (Yadav et al., 2023). Concretely, after RLER training, we merge $\{\theta_k\}_{k=1}^{K}$ into a single set of weights $\theta_{\mathrm{merge}}$ and use $\pi_{\theta_{\mathrm{merge}}}$ for deployment. We use the default TIES hyperparameters ($\kappa = 0.7$, $\alpha = 0.5$), which we find robust across settings.

## 5 Experiments

We design our experiments to answer two questions: (i) can RLER effectively resolve the systemic biases diagnosed in §3, and (ii) can it deliver stable performance gains when scaling on unlabeled data? In §5.2, we benchmark RLER against RLIR and RLVR baselines, quantitatively validating its improvements in **Accuracy**, **Unbiasedness**, and **Robustness** via the three diagnostic metrics ($\rho_{\mathrm{noise}}$, $\rho_{\mathrm{selfbias}}$, $\rho_{\mathrm{symbias}}$). In §5.3, we conduct fine-grained ablations to isolate the contributions of *Ensemble Rewarding*, *Adaptive Interpolation*, and *Rollout Selection*, and analyze their specific roles in mitigating each type of bias. Finally, §5.4 demonstrates the practical value of RLER as a stably scaling solution for unlabeled reinforcement learning; extended experiments on cross-model and cross-dataset generalization, hyperparameters robustness, and ensemble-size scaling with compute analysis are reported in the appendix B.

### 5.1 Experimental Settings

**Models.** Our main testbed follows the standard RLVR/R-LIR setup and uses the Qwen2.5 series (Yang et al., 2024b;a), with QWEN2.5-MATH-7B as the default backbone. For cross-model generalization, we also evaluate RLER on LLAMA-3.2-3B-INSTRUCT, LLAMA-3.1-8B-INSTRUCT (Grattafiori et al., 2024), and QWEN2.5-7B-INSTRUCT (see Appendix B.1).

**Datasets and Benchmarks.** For our main analyses, we consider two math-style, verifiable reasoning corpora: (i) an arithmetic dataset (with a 500-problem in-distribution test split), and (ii) DAPO-MATH-17K (Yu et al., 2025). On DAPO-MATH-17K, we train QWEN2.5-MATH-7B and evaluate on six challenging benchmarks: MATH500 (Hendrycks et al., 2021), AMC23 (Li et al., 2024), AMC24, AIME24 (Li et al., 2024), AIME25 (MAA, 2024), and HMMT24. For cross-model and cross-dataset generalization, we additionally use LLAMA-3.2-3B-INSTRUCT on the arithmetic dataset, LLAMA-3.1-8B-INSTRUCT on BIG-MATH (Albalak et al., 2025), and QWEN2.5-7B-INSTRUCT on WebInstruct-verified (Ma et al., 2025) with evaluation on MMLU-PRO (Wang et al., 2024). We report both Avg@k and Pass@k across all settings.

**Baselines.** We compare RLER against both RLIR and RLVR methods. For RLIR, we include representa-

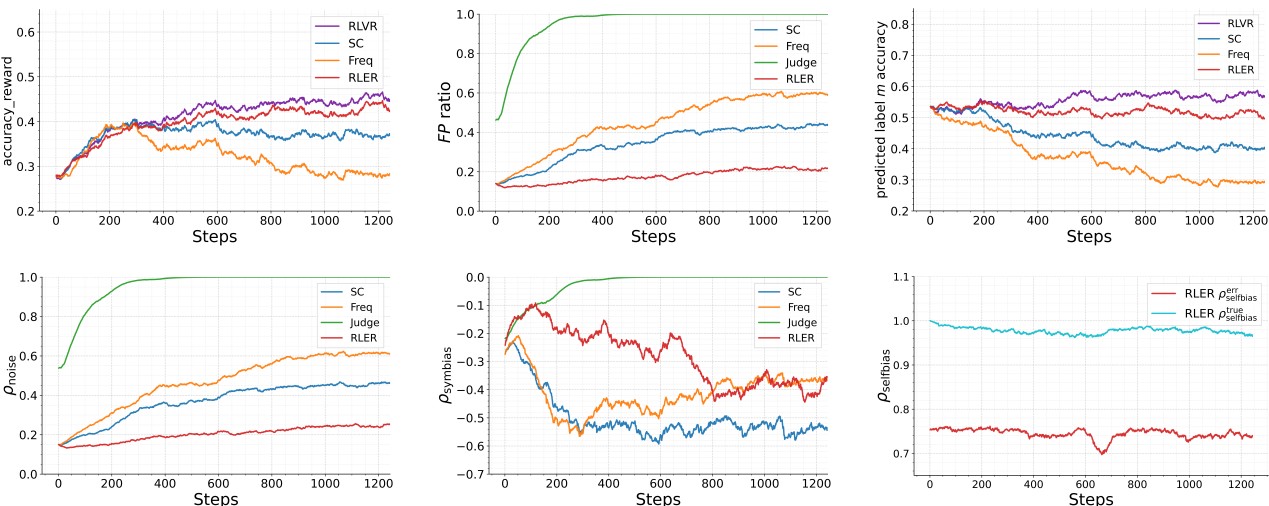

*Figure 4.* Training dynamics on DAPO-MATH-17K. Compared with RLIR baselines, RLER achieves steadier accuracy improvement while reducing reward noise, over-reward skew, and erroneous policy–reward coupling, closely tracking the RLVR upper bound.

tive hard- and soft-reward paradigms: hard-reward Self-Consistency (SC) and LLM-as-a-Judge (Judge), the soft-reward frequency-based approach (Freq), and recent stronger RLIR baselines including INTUITOR (Zhao et al., 2025) and Co-rewarding (Zhang et al., 2025c). For RLVR, we adopt an oracle-labeled setting with exact answer checking as an upper bound.

**Details.** All methods are implemented in the Open-R1 framework and trained with GRPO. For DAPO-MATH-17K, we fix the rollout budget per query to $G{=}16$. In RLER, we use an ensemble of $k{=}2$ sub-policies by default, so that each sub-policy generates $G_k{=}8$ rollouts and the total rollout budget matches single-model RLIR. An ensemble-size scaling study with compute and memory analysis in Appendix B.3 shows that $k{=}2$ offers the best trade-off under a fixed rollout budget. Unless otherwise specified, we use a learning rate of $1{\times}10^{-6}$, KL regularization coefficient $\beta{=}0.001$, and sampling temperature 0.9. Further training details and additional results are provided in Appendix B, and full prompt templates are given in Appendix D.

### 5.2 Main Results

**Accuracy.** The results on DAPO-MATH-17K and its evaluation benchmarks are shown in Figure 4 and Table 1. In terms of performance, RLER consistently outperforms both classical RLIR baselines and recent stronger RLIR methods. RLER recovers about 96.0% of the test accuracy of RLVR, corresponding to an average gain of +45.9% over the pretrained model and +6.2% over the best RLIR baseline. To explain these gaps, we analyze the diagnostic metrics from §3. As shown in Figure 4, RLER substantially reduces $\rho_{\text{noise}}$ throughout training so that accuracy rises steadily and closely tracks the RLVR curve, whereas standard RLIR

methods accumulate noise and eventually plateau. Beyond this main configuration, we observe similar relative improvements of RLER over RLIR baselines, and performance close to the RLVR upper bound, across additional backbones and datasets; full results are reported in Appendix B.

**Unbiasedness.** To examine unbiasedness, we focus on $\rho_{\text{symbias}}$, which measures the imbalance between over-reward (FP) and under-reward (FN) errors. RLIR baselines exhibit a strong over-reward skew: most reward noise comes from false positives, and in the decoupling experiment, we show that Over-reward is more detrimental than under-reward. By contrast, RLER markedly suppresses the FP-dominated component of $\rho_{\text{noise}}$ and drives $\rho_{\text{symbias}}$ toward a much more symmetric regime. This effect mainly comes from the rollout selection, which aggressively filters high-confidence FPs, together with the reward interpolation that simultaneously dampens residual FP rewards and recovers under-rewarded FNs within the unified ensemble reward space. Consequently, the reward noise no longer induces a systematic drift toward over-reward, preventing the early "over-reward bias amplification" regime and effectively raising the performance ceiling.

**Robustness.** As discussed in Finding 4, robustness requires that reward estimates do not tightly follow the policy's own confidence, especially on erroneous predictions; otherwise self-rewarding RL quickly falls into a self-confirming loop. We therefore track the policy–reward coupling metric $\rho_{\text{selfbias}}$, decomposed into $\rho_{\text{selfbias}}^{\text{true}}$ and $\rho_{\text{selfbias}}^{\text{err}}$. RLIR baselines exhibit high coupling in both regimes, meaning that the reward system strongly reinforces whatever the policy is most confident in, regardless of correctness. In contrast, RLER maintains $\rho_{\text{selfbias}}^{\text{true}} \approx 1$ while substantially

*Table 1.* Main results on DAPO-MATH-17K with QWEN2.5-MATH-7B. We report Avg@8 on six reasoning benchmarks and Pass@8 in the final column. RLER achieves the best overall Avg@8 and Pass@8 among RLIR methods, outperforming both classical and recent stronger baselines.

| Method | Benchmarks | | | | | | Overall | |
|---|---|---|---|---|---|---|---|---|
| | AIME24 | AIME25 | AMC23 | AMC24 | MATH500 | HMMT24 | Avg@8 | Pass@8 |
| Pre-RL | 12.5 | 6.4 | 45.3 | 23.0 | 59.2 | 7.9 | 25.7 | 54.8 |
| *RLVR* | *32.1* | *12.5* | *65.0* | *34.2* | *79.1* | *10.4* | *38.9* | *55.5* |
| *RLIR* | | | | | | | | |
| Judge | 3.3 | 1.7 | 23.1 | 18.4 | 34.1 | 0.0 | 13.4 | 22.5 |
| SC | 16.3 | **13.8** | 55.9 | 32.8 | 75.0 | 4.2 | 33.0 | 47.1 |
| Freq | 11.7 | 8.8 | 43.1 | 25.8 | 71.7 | 1.7 | 27.1 | 31.6 |
| INTUITOR | 21.7 | 13.3 | 57.2 | 31.1 | 73.8 | 5.0 | 33.7 | 48.3 |
| Co-rewarding | **23.3** | 11.7 | 60.0 | 31.4 | 74.5 | **11.3** | 35.3 | 51.2 |
| **RLER** | **23.3** | 12.1 | **66.9** | **35.8** | **77.5** | 9.6 | **37.5** | **52.8** |

suppressing $\rho_{\text{selfbias}}^{\text{err}}$, indicating that confidence is trusted only when the answer is actually correct. Externally, this decoupling manifests as much lower training-time accuracy variance and lower prediction entropy across checkpoints and sampling temperatures, while Maj@k and Pass@k remain stable or improve (see Table B.2.1 and Figure 4 for details). Taken together, these results show that RLER constructs a reward space that is robust to policy drift and sampling noise, avoiding the self-confirming instability mode characteristic of standard RLIR.

### 5.3 Variants Ablations

**Ensemble-based Unified Rewarding.** We assess the contribution of each component by ablating *Model Merge*, *Rollout Selection*, *Reward Interpolation*, and *Ensemble* from RLER individually. As shown in Figure 5, the pronounced degradation when removing the *Ensemble* indicates that mitigating system bias to improve accuracy is the most critical factor. Furthermore, we find that performing *Reward Interpolation* within the ensemble space yields superior performance. We hypothesize that this stems from the ensemble's unified reward space: diversity across models reduces $\rho_{\text{selfbias}}^{\text{err}}$, improves the robustness of the reward space, and enables $\alpha(x)$ to be estimated more accurately and stably within the ensemble space.

**Adaptive Soft-reward Interpolation.** As shown in Figure 5, removing *Reward Interpolation* leads to a substantial performance drop. We further analyze the necessity of each component in our interpolation method: starting from *Int v3* (ours), dropping the calibrated mass $s_k$ and the *Batch-wise Min-Max Normalization* $\hat{c}_k$ yields *Int v2*; further removing the confidence estimate $\bar{c}_k(j)$ produces *Int v1*, where we instead control the interpolation strength via annealing (with $\alpha$ decaying over training steps). We measure the interpolation gain as $|r^{\text{H}} - r^{\star}| - |r^{(\alpha)} - r^{\star}|$. The results show that Int

v3 attains the best performance and the largest interpolation gain, confirming the contribution of each step.

*Table 2.* Disagreement-aware rollout selection analysis. We compare selection rules by the average number of selected rollouts $b_{\text{avg}}$ and reward noise $\rho_{\text{noise}}$ under correct ($m = t$) and incorrect ($m \neq t$) ensemble majority predictions.

| Method | $m = t$ | $m \neq t$ | |
|---|---|---|---|
| | $b_{\text{avg}}$ | $\rho_{\text{noise}}$ | $b_{\text{avg}}$ |
| select all | 16.0 | 65.5 | 16.0 |
| $m$ only | 12.1 | 100.0 | 9.9 |
| $m$ except | 3.9 | **8.7** | 6.1 |
| **RLER** | **12.0** | 50.5 | **11.3** |

**Disagreement-Aware Rollout Selection.** We evaluate *Rollout Allocation* and *Rollout Selection* in Figure 6 and Table 2. For allocation, we measure diversity gain as the accuracy gap between the ensemble $m^{\text{EC}}$ and the average individual model, $\Delta_{\text{div}} = \text{Acc}(m^{\text{EC}}) - \frac{1}{M}\sum_{i=1}^{M}\text{Acc}(m_i)$. *Data Sharding* yields a larger $\Delta_{\text{div}}$, indicating that distributing training data across sub-policies effectively increases ensemble diversity. For selection, we compare strategies using the average number of selected rollouts $b_{\text{avg}}$ and the reward noise rate $\rho_{\text{noise}}$ conditioned on whether $m^{\text{EC}}$ is correct. Here, *m only* selects only $m^{\text{EC}}$, while *m except* excludes $m^{\text{EC}}$. Our disagreement-aware strategy selects substantially more rollouts when $m^{\text{EC}} = t$ and aggressively filters FP rollouts when $m^{\text{EC}} \neq t$, achieving lower $\rho_{\text{noise}}$ than *select all*. Thus, *Rollout Selection* uses ensemble disagreement to both improve accuracy and reshape the noise away from FP-dominated over-reward, contributing to lower $\rho_{\text{symbias}}$.

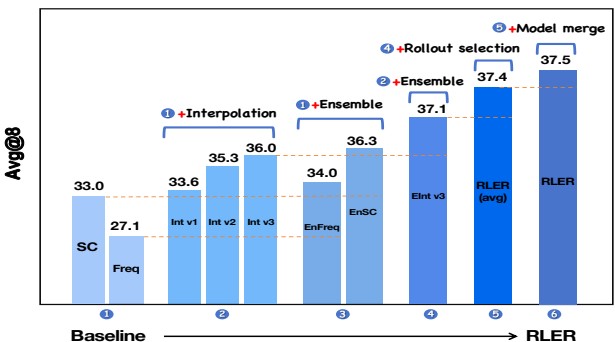
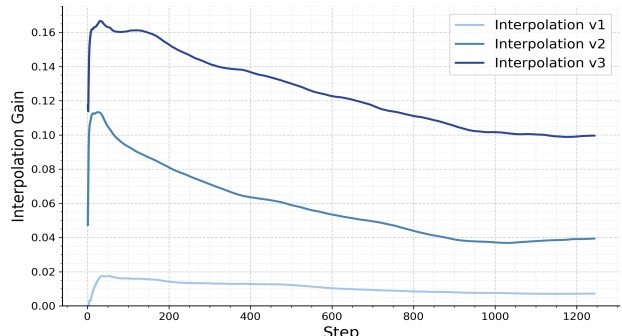

*Figure 5.* Component ablations of RLER. Left: removing ensemble rewarding, adaptive interpolation, rollout selection, or model merging reduces Avg@8, showing that the components contribute cumulatively. Right: the full adaptive interpolation design yields the largest reward-improvement margin over hard rewards.

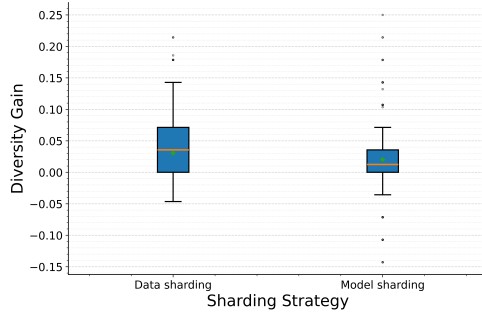

*Figure 6.* Rollout allocation and ensemble diversity. We compare allocation strategies using the diversity gain $\Delta_{\mathrm{div}}$, defined as the accuracy improvement of the ensemble majority over the average individual policy prediction.

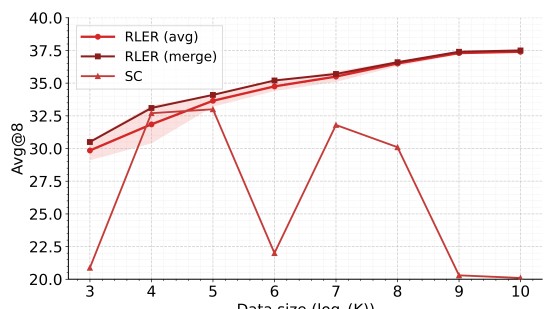

*Figure 7.* Unlabeled-data scaling on the test benchmarks. RLER maintains smooth Avg@8 improvement as the training set grows, while RLIR baseline show less stable scaling behavior.

## 5.4 Practical Value of RLER

**Stably Scalable Unlabeled RL.** In real-world deployments, labels are scarce while unlabeled data and compute are limited, so practitioners cannot know in advance how much data is needed to reach optimal performance. A key requirement is therefore that an unlabeled RL algorithm remain *stably scalable* as more data is added. To assess the practical value of RLER, we examine performance as a function of training set size, as shown in Figure 7. Compared with RLIR baselines, RLER shows smooth, consistently improving behavior from 8k up to 1024k samples. Notably, the merged model not only resolves the multi-model deployment issue but also attains comparable or higher accuracy with reduced variance, making RLER a practical choice for large-scale unlabeled RL.

## 6 Conclusions

We investigated why self-rewarding RL (RLIR) remains less stable and effective than RLVR, and traced this gap to a systemic reward bias, quantified by three diagnostic metrics for noise, self-bias, and asymmetric over-reward. Based

on this diagnosis, we proposed RLER, which aggregates diverse policies into a unified reward-estimation space and uses adaptive soft-reward interpolation with disagreement-aware rollout selection to improve accuracy, unbiasedness, and robustness. Empirically, RLER outperforms strong RLIR baselines, closely approaches RLVR, and exhibits smooth, stable scaling on large unlabeled corpora under a practical compute.

## Acknowledgements

This work is supported by the Beijing Natural Science Foundation (Grant Nos. 4262065, 4222037, and L181010).

## Impact Statement

This paper presents work whose goal is to advance the field of Machine Learning. There are many potential societal consequences of our work, none which we feel must be specifically highlighted here.

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

# A Algorithmic Workflow of RLER

---

**Algorithm 1** RLER: Ensemble-Based Unified Rewarding with Adaptive Interpolation and Disagreement-Aware Selection

---

1: **Input:** unlabeled query set $\mathcal{D}_u$; source policies $\{\pi_{\theta_k}\}_{k=1}^K$; per-policy rollout budgets $\{G_k\}_{k=1}^K$; confidence-calibration bounds $(\beta^-, \beta^+)$; training steps $T$
2: **Output:** merged deployable policy $\pi_{\theta_{\mathrm{merge}}}$
3: **for** $t = 1$ to $T$ **do**
4:     Sample a batch of queries $\mathcal{B} \subset \mathcal{D}_u$ and initialize $\mathcal{S} \leftarrow \emptyset$.
5:     **for** each query $x \in \mathcal{B}$ **do**
6:         **Source-local rollout statistics:**
7:         **for** $k = 1$ to $K$ **do**
8:             Draw $\mathcal{Y}_k(x) = \{y_{k,i}\}_{i=1}^{G_k} \sim \pi_{\theta_k}(\cdot \mid x)$.
9:             Compute answer distribution $p^{(k)}(x)$ and calibrated masses $s_k(j)$.
10:         **end for**
11:         **Unified reward construction:**
12:         Pool rollouts $\mathcal{Y}(x) = \bigcup_{k=1}^K \mathcal{Y}_k(x)$.
13:         Compute $\bar{p}_j(x) = \frac{1}{K} \sum_{k=1}^K p_j^{(k)}(x)$ and $m^{\mathrm{EC}}(x) = \arg\max_j \bar{p}_j(x)$.
14:         Compute $\tilde{p}_j(x) = \frac{1}{K} \sum_{k=1}^K s_k(j)$ and $\alpha(x) = \mathrm{clip}(\tilde{p}_{m^{\mathrm{EC}}}(x), 0, 1)$.
15:         **for** each rollout $y_i \in \mathcal{Y}(x)$ **do**
16:             Set $r_i^{\mathrm{H}} = \mathbf{1}[\ell(y_i) = m^{\mathrm{EC}}(x)]$, $r_i^{\mathrm{S}} = \bar{p}_{\ell(y_i)}(x)$.
17:             Set $r_i^{(\alpha)} = (1 - \alpha(x)) r_i^{\mathrm{S}} + \alpha(x) r_i^{\mathrm{H}}$.
18:         **end for**
19:         **Disagreement-aware rollout selection:**
20:         Group rollouts by final answer label and set $w_{m^{\mathrm{EC}}}(x) = \alpha(x)$, $w_j(x) = 1 - \tilde{p}_j(x)$ for $j \neq m^{\mathrm{EC}}(x)$.
21:         Set per-answer quota $G$ using the per-policy rollout budget as the cap.
22:         Compute $\mathrm{take}_y = \min\{G, \mathrm{round}(G \cdot w_y(x))\}$ and $b(x) = \sum_y \mathrm{take}_y$.
23:         Add selected rollouts and rewards to $\mathcal{S}$.
24:     **end for**
25:     **for** $k = 1$ to $K$ **do**
26:         Update $\pi_{\theta_k}$ by GRPO on the subset of $\mathcal{S}$ assigned to source $k$.
27:     **end for**
28: **end for**
29: Merge source policies with TIES-Merging to obtain $\pi_{\theta_{\mathrm{merge}}}$.

---

# B More Experiment Details

## B.1 Cross-Model and Cross-Dataset Generalization

**Why math-style, verifiable reasoning tasks.** We adopt math and competition-style problems as our main testbed because their discrete, verifiable answers provide accurate oracle rewards, matching the dominant RLVR/RLIR setup. This setting allows us (i) to precisely compute $\rho_{\mathrm{noise}}, \rho_{\mathrm{selfbias}}, \rho_{\mathrm{symbias}}$ and run controlled decoupling studies, and (ii) to fairly compare RLER with RLIR baselines against an RLVR upper bound under clean, well-understood conditions, before moving to broader cross-model and cross-dataset generalization.

**Extension beyond math-style tasks.** Conceptually, RLER only assumes that textual answers can be reliably scored for semantic equivalence. We therefore also train RLER with QWEN2.5-7B-INSTRUCT on WebInstruct-verified using the official verifier, and evaluate on MMLU-PRO against strong RLIR baselines including INTUITOR (Zhao et al., 2025). Across these non-mathematical, multi-domain tasks, RLER again improves over RLIR and remains close to RLVR, providing additional evidence for its cross-model and cross-dataset generalization beyond math-style benchmarks.

### B.1.1 RLER ON ARITHMETIC DATASET

**Experimental settings.** Prior work shows that RL gains are highly sensitive to model pretraining: pretraining on large-scale web corpora can introduce data contamination on popular benchmarks (Wu et al., 2025; Shao et al., 2025). To eliminate contamination effects and cleanly validate our method, we synthesize a decontaminated arithmetic dataset (375k) comprising expressions over operators $\{+, -, //, \%\}$, with $1-3$ operators applied to $2-6$-digit integers, partitioned into 15 uniformly distributed difficulty groups with increasing hardness. We evaluate on a 500-problem in-distribution test set.

We consider two instruction-tuned backbones: QWEN-2.5-1.5B-INSTRUCT and LLAMA-3.2-3B-INSTRUCT. Unless otherwise stated, the RL setup follows the main configuration in §5.1. We set the rollout budget to $G{=}32$, learning rate to $1{\times}10^{-6}$, KL regularization coefficient to $\beta{=}0.001$, sampling temperature to $0.9$, and train for one epoch. RLVR uses oracle exact-answer checking as the verifiable upper bound, while RLIR baselines include FREQ, SC, JUDGE, and INTUITOR(Zhao et al., 2025). All experiments are conducted on NVIDIA H20 (96 GB).

**Main results.** The results of compared methods for QWEN-2.5-1.5B-INSTRUCT are reported in Table 3, the ablation results for RLER are shown in Table 5, and those for LLAMA-3.2-3B-INSTRUCT are given in Table 4. In both cases, RLER achieves the best overall performance among RLIR methods and substantially closes the gap to RLVR. For Qwen, RLER improves Avg@16 by +14.1 points over the best RLIR baseline and incurs the smallest Pass@16 degradation relative to the pre-RL model. For Llama, RLER consistently outperforms all RLIR baselines across ensemble configurations, with the best variant ($k{=}4, G_k{=}8$) achieving strong gains on both Avg@16 and Pass@16. These consistent trends across architectures support that RLER's bias-mitigation effect is not tied to a particular backbone. The ablations on QWEN-2.5-1.5B-INSTRUCT (Table 5) further show that removing the ensemble, interpolation, or rollout selection consistently harms performance, indicating that all three components contribute cumulatively to RLER's gains.

To further rule out contamination or memorization as an explanation of these gains, we additionally run RLIR on the arithmetic dataset under a random-reward regime. In this setting, model fails to improve over the pre-RL baseline and in fact often degrade (see Figure 8), confirming that RLER's improvements on the decontaminated arithmetic corpus require informative reward signals rather than hidden data leakage.

### B.1.2 RLER ON BIGMATH

We further examine cross-model and cross-dataset generalization on BIGMATH (Albalak et al., 2025) with LLAMA-3.1-8B-INSTRUCT, using the same RL setup and evaluation protocol as in the main DAPO-MATH-17K experiments. We compare pre-RL, RLVR, RLIR baselines (SC, Judge, INTUITOR), and RLER under the default ensemble configuration ($k{=}2, G_k{=}8$).

As shown in Table 6, RLER consistently improves over all RLIR baselines on both Avg@8 and Pass@8, while substantially narrowing the gap to RLVR. In particular, RLER outperforms the strongest RLIR baseline (INTUITOR) by +1.72 Avg@8 and +3.35 Pass@8, while remaining within 1.2 Avg@8 and 2.3 Pass@8 of RLVR, confirming that our bias-mitigation strategy transfers to a harder corpus and a different backbone.

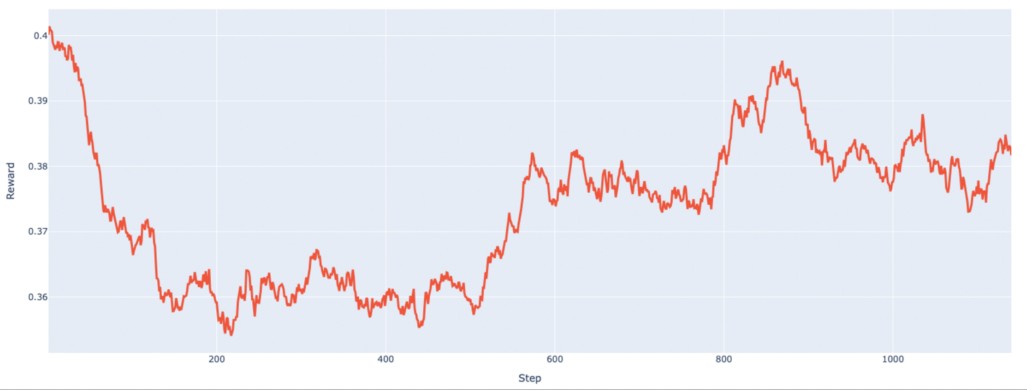

*Figure 8.* Training curve of Self-Consistency (SC) on the arithmetic dataset with random rewards.The x-axis is the training step and the y-axis is accuracy reward. The model(Qwen-2.5-1.5B-Instruc) fails to improve over the pre-RL baseline and sometimes even degrades.

*Table 3.* Zero-shot Avg@16 and Pass@16 of QWEN-2.5-1.5B-INSTRUCT on the arithmetic test set.

| Method | Avg@16 | Pass@16 |
|---|---|---|
| pre-RL | 41.5 | 89.2 |
| *RLVR* | 93.2 | 95.5 |
| *RLIR* | | |
| Judge | 48.3 | 70.6 |
| SC | 57.4 | 60.2 |
| Freq | 56.9 | 62.6 |
| INTUITOR | 60.4 | 61.5 |
| *RLER* ($k{=}2, G_k{=}8$) | 69.2 | 72.2 |
| *RLER* ($k{=}4, G_k{=}8$) | 71.5 | 75.8 |

*Table 4.* Zero-shot Avg@16 and Pass@16 of LLAMA-3.2-3B-INSTRUCT on the arithmetic test set.

| Method | Avg@16 | Pass@16 |
|---|---|---|
| pre-RL | 30.7 | 80.2 |
| *RLVR* | 70.2 | 87.8 |
| *RLIR* | | |
| Freq | 47.5 | 56.2 |
| SC | 48.4 | 54.4 |
| Judge | 41.7 | 77.3 |
| INTUITOR | 50.3 | 54.2 |
| *RLER* ($k{=}2, G_k{=}8$) | 58.7 | 69.8 |
| *RLER* ($k{=}4, G_k{=}4$) | 62.3 | 74.4 |
| *RLER* ($k{=}8, G_k{=}2$) | 61.9 | 78.6 |
| *RLER* ($k{=}4, G_k{=}8$) | 63.9 | 75.2 |

### B.1.3 RLER ON WEBINSTRUCT-VERIFIED

Conceptually, RLER only requires that textual answers can be reliably compared in a common reward space, i.e., a verifier that can quantify semantic equivalence between different responses to the same query. To test this in a non-mathematical, open-domain setting, we train QWEN2.5-7B-INSTRUCT on WEBINSTRUCT-VERIFIED, a diverse, high-quality corpus with an officially provided LLM verifier, and evaluate on MMLU-PRO, a challenging multi-task benchmark (12K questions across 14 disciplines). The verifier scores correctness by comparing model outputs against reference answers, and these scores are used as rewards. We compare pre-RL, RLVR, RLIR baselines (SC, Judge, INTUITOR), and RLER with $k{=}2$, $G_k{=}8$.

Table 7 reports Pass@1 for five representative categories and the average over all 14 categories. RLER consistently improves over all RLIR baselines and substantially narrows the gap to RLVR, indicating that our bias-mitigation strategy extends beyond math-style tasks to broader, open-domain reasoning as long as a reasonably accurate verifier is available.

### B.2 Hyperparameter Robustness

Although the reward design of RLER may appear complex, the number of method-specific hyperparameters that actually require manual tuning is small. In practice, only two quantities are exposed to the user: (i) the ensemble size $k$ (analyzed in Appendix. B.3), and (ii) the batch-level interpolation bounds $\beta^-, \beta^+$ used in Adaptive Soft-Reward Interpolation. All other components (e.g., confidence normalization and selection weights) are fully data-driven and computed adaptively from current-batch statistics. We investigate the robustness of RLER to two key hyperparameters: (i) the sampling temperature used for rollout generation, and (ii) the confidence bounds $\beta^-, \beta^+$ in Adaptive Soft-Reward Interpolation. Overall, RLER maintains consistent gains over RLIR baselines, with reduced performance variance across all tested settings.

*Table 5.* Ablation study of RLER on the arithmetic test set with QWEN-2.5-1.5B-INSTRUCT. We report Avg@16 on the test set when progressively removing *Ensemble*, *Adaptive Interpolation*, and *Disagreement-Aware Rollout Selection*."w/o all" reduces to single-model RLIR baselines (SC/Freq).

| Method | Avg@16 |
|---|---|
| *RLER* | 71.5 |
| *w/o Rollout selection* | |
| Ensemble Interpolation v3 | 69.6 |
| Ensemble Interpolation v2 | 68.2 |
| *w/o Interpolation&Rollout selection* | |
| Ensemble SC | 67.6 |
| Ensemble Freq | 65.8 |
| *w/o Ensemble&Rollout selection* | |
| SC Interpolation v3 | 63.2 |
| SC Interpolation v2 | 61.3 |
| SC Interpolation v1 | 59.8 |
| *w/o all* | |
| SC | 57.4 |
| Freq | 56.9 |

*Table 6.* Results on BIGMATH with LLAMA-3.1-8B-INSTRUCT. We report per-subset accuracy and aggregate Avg@8 / Pass@8.

| Method | MATH500 | AIME24 | AIME25 | AMC23 | AMC24 | HMMT24 | Avg@8 | Pass@8 |
|---|---|---|---|---|---|---|---|---|
| pre-RL | 45.75 | 3.75 | 0 | 19.38 | 12.22 | 0 | 13.52 | 30.20 |
| *RLVR* | 49.95 | 5.00 | 3.33 | 26.25 | 18.13 | 1.25 | 17.32 | 37.31 |
| *RLIR* | | | | | | | | |
| SC | 42.65 | 2.50 | 2.08 | 24.37 | 12.78 | 0 | 14.06 | 30.85 |
| Judge | 37.22 | 0.42 | 1.25 | 15.62 | 10.83 | 0 | 10.89 | 25.39 |
| INTUITOR | 46.25 | 3.33 | 2.50 | 21.56 | 12.78 | 0.42 | 14.47 | 31.72 |
| *RLER* ($k=2$, $G_k=8$) | 48.35 | 4.17 | 2.92 | 23.75 | 16.67 | 1.25 | 16.19 | 35.07 |

### B.2.1 SAMPLING TEMPERATURE

We compare Self-Consistency (SC) and RLER under different sampling temperatures $t \in \{0.5, 0.7, 0.9, 1.0\}$ on two setups: LLAMA-3.2-3B-INSTRUCT on the arithmetic corpus and QWEN2.5-MATH-7B on DAPO-MATH-17K. For each setting, we report: Pass@1, checkpoints accuracy variance (across $\pm 5$ checkpoints around the best one), and average rollouts entropy.

Across all temperatures and both models, RLER:

1. consistently improves Pass@1 over SC;

2. dramatically reduces checkpoint accuracy variance (from tens of points to $\approx 1$–4), indicating more stable training;

3. yields lower prediction entropy, suggesting more calibrated, less noisy reward signals.

These trends support that RLER reduces, rather than amplifies, sensitivity to sampling temperature.

### B.2.2 INTERPOLATION BOUNDS IN ADAPTIVE SOFT-REWARD INTERPOLATION

In RLER, Adaptive Soft-Reward Interpolation uses batch-level confidence ranges $[\beta^-, \beta^+]$ to modulate the hard/soft reward mixture for each source model: items above $\beta^+$ rely more on hard rewards, whereas items below $\beta^-$ receive stronger soft-reward smoothing.

*Table 7.* Pass@1 on MMLU-PRO for models trained on WEBINSTRUCT-VERIFIED with QWEN2.5-7B-INSTRUCT. We show five representative categories and the average over all 14 categories.

| Method | biology | business | chemistry | comp. sci. | engineering | Pass@1 (avg) |
|---|---|---|---|---|---|---|
| pre-RL | 59.55 | 59.70 | 48.23 | 52.44 | 34.47 | 48.72 |
| *RLVR* | 72.11 | 62.96 | 51.86 | 57.34 | 42.39 | 56.37 |
| *RLIR* | | | | | | |
| SC | 70.29 | 60.71 | 53.27 | 56.10 | 40.56 | 54.05 |
| Judge | 70.71 | 61.72 | 49.56 | 57.56 | 37.46 | 52.82 |
| INTUITOR | 70.05 | 61.23 | 51.11 | 56.32 | 39.87 | 53.06 |
| Co-rewarding | 70.25 | 61.45 | 51.87 | 56.13 | 40.27 | 54.82 |
| *RLER* ($k=2$, $G_k=8$) | 70.83 | 61.59 | 52.22 | 56.28 | 41.30 | 55.12 |

*Table 8.* Temperature robustness of SC and RLER on the arithmetic dataset (LLAMA-3.2-3B-INSTRUCT) and DAPO-MATH-17K (QWEN2.5-MATH-7B). Each cell shows *Pass@1 / checkpoints accuracy variance (across $\pm 5$ checkpoints around the best one) / average rollouts entropy*.

| Model | Method | $t = 0.5$ | $t = 0.7$ | $t = 0.9$ | $t = 1.0$ |
|---|---|---|---|---|---|
| Llama-3.2-3B-Instruct | SC | 51.9 / 78.92 / 1.09 | 51.3 / 33.90 / 1.83 | 48.4 / 3.44 / 1.90 | 49.3 / 6.77 / 3.66 |
| (Arithmetic dataset) | RLER | 57.2 / 4.26 / 0.70 | 58.3 / 2.78 / 0.92 | 58.7 / 2.50 / 1.39 | 59.6 / 3.70 / 2.37 |
| Qwen2.5-Math-7B | SC | 32.4 / 72.50 / 0.08 | 31.8 / 38.00 / 0.10 | 33.0 / 32.11 / 0.17 | 32.7 / 34.90 / 0.32 |
| (DAPO-Math-17K) | RLER | 37.6 / 2.10 / 0.05 | 36.9 / 1.23 / 0.09 | 37.5 / 2.10 / 0.06 | 38.8 / 1.23 / 0.09 |

To assess sensitivity, we vary $\beta^- \in \{0.10, 0.20, 0.30\}$ and $\beta^+ \in \{0.50, 0.60, 0.70\}$, and report Avg@8 on the six math benchmarks with QWEN2.5-MATH-7B:

*Table 9.* Sensitivity of RLER to interpolation bounds $\beta^-$ and $\beta^+$. We report Avg@8 on the six math benchmarks.

| $\beta^-$ | $\beta^+$ | | |
|---|---|---|---|
| | 0.50 | 0.60 (default) | 0.70 |
| 0.10 | 37.0 | 37.3 | 37.1 |
| 0.20 (default) | 37.4 | 37.5 | 37.3 |
| 0.30 | 36.9 | 37.2 | 37.0 |

Performance varies only within a narrow band (about $\pm 0.2$–$0.3$ around the default), and $\beta^-=0.20$, $\beta^+=0.60$ is consistently near optimal. In practice, we find that $\beta^- \in [0.10, 0.40]$ and $\beta^+ \in [0.50, 0.80]$ all yield very similar performance, indicating that RLER is not highly sensitive to the exact choice of these bounds.

**Fixed versus adaptive interpolation.** To further isolate the role of the sample-dependent gate $\alpha(x)$, we compare fixed interpolation weights with progressively more adaptive variants. As shown in Table 10, fixed interpolation improves over naive hard rewards only modestly, while the full adaptive gate achieves the best performance both in the single-policy interpolation setting and in the full RLER framework. This confirms that the gain is not merely due to adding an interpolation coefficient, but comes from query-dependent weighting based on ensemble reliability.

### B.3 Ensemble Size Scaling and Compute Cost

**Diversity evolution during training.** All RLER source policies start from the same backbone initialization; their diversity emerges from data sharding, independent sampling, and separate updates. To directly examine this process, we track three diversity measures throughout training under the main DAPO-MATH-17K setting: answer-distribution JSD, rollout semantic diversity, and the ensemble diversity gain $\Delta_{\text{div}}$. Table 11 shows that sub-policies rapidly decorrelate after training begins, and that this decorrelation translates into a measurable ensemble benefit.

*Table 10.* Fixed versus adaptive reward interpolation on DAPO-MATH-17K with QWEN2.5-MATH-7B. We report Avg@8 over the six math benchmarks. Adaptive $\alpha(x)$ consistently outperforms fixed or schedule-based interpolation.

| Method | Interpolation / Gate | Avg@8 |
|---|---|---|
| Int fix | Fixed $\alpha = 0.5$ | 32.7 |
| Int fix | Fixed $\alpha = 0.7$ | 33.5 |
| Int v1 | Annealed schedule | 33.6 |
| Int v2 | Simplified adaptive gate | 35.3 |
| Int v3 | Full adaptive $\alpha(x)$ | 36.0 |
| RLER fix | Fixed $\alpha = 0.5$ + full RLER | 36.2 |
| RLER fix | Fixed $\alpha = 0.7$ + full RLER | 36.6 |
| RLER | Full adaptive $\alpha(x)$ | **37.5** |

*Table 11.* Diversity evolution of RLER source policies during training. Even with the same initialization, data sharding and separate updates quickly increase answer-level and trajectory-level diversity, yielding positive ensemble diversity gain $\Delta_{\text{div}}$.

| Training progress | 0% | 20% | 40% | 60% | 80% | 100% |
|---|---|---|---|---|---|---|
| Answer-distribution JSD | 0.0141 | 0.0708 | 0.0794 | 0.0706 | 0.0687 | 0.0683 |
| Rollout semantic diversity | 0.0258 | 0.0873 | 0.0856 | 0.0834 | 0.0889 | 0.0842 |
| $\Delta_{\text{div}}$ | 0.0054 | 0.0373 | 0.0493 | 0.0400 | 0.0388 | 0.0206 |

**Main-setting compute cost.** Under our main experimental setting, we fix the total rollout budget per query to $G_{\text{total}} = 16$ (Sec. 5.1): single-model RLIR uses one policy to generate 16 rollouts, while RLER with $k=2$ uses two sub-policies that each generate $G_k=8$ rollouts, keeping the total budget unchanged. To compare compute more precisely, we decompose one RL step into: (i) rollout generation; (ii) reward/advantage computation; and (iii) policy update. Rollout selection is applied after (ii), so the dominant compute difference comes from (iii), which scales with the number of rollouts that actually enter the loss.

Let $b_{\text{avg}}$ be the average number of selected rollouts per query, $\gamma \approx 2$ be the backward/forward FLOPs ratio, and $G_{\text{base}}$ be the baseline rollout count (here $G_{\text{base}}=16$). We define a relative compute proxy

$$\tilde{C}_{\text{rel}} = \frac{G_{\text{total}} + \gamma\, b_{\text{avg}}}{G_{\text{base}} + \gamma\, G_{\text{base}}}. \tag{15}$$

Table 12 summarizes the main-setting values. Under the same rollout budget, RLER has slightly lower update FLOPs than RLIR; the only substantial extra cost is memory, since loading $k$ sub-models in parallel increases VRAM approximately linearly in $k$.

*Table 12.* Relative compute under the main setting on DAPO-MATH-17K with QWEN2.5-MATH-7B.

| Method | $k$ | $G_{\text{total}}$ | $b_{\text{avg}}$ | $\tilde{C}_{\text{rel}}$ |
|---|---|---|---|---|
| RLIR (single-model) | 1 | 16 | 16.0 | 1.00 |
| RLER (ours) | 2 | 16 | 12.0 | 0.83 |

**Ensemble-size scaling.** We further study how ensemble size $k$ trades off accuracy, diversity, and cost. Table 13 reports Avg@8 / Pass@8, the diversity gain $\Delta_{\text{div}} = \text{Acc}(m^{\text{EC}}) - \frac{1}{M}\sum_{i=1}^{M} \text{Acc}(m_i)$, the same compute proxy $\tilde{C}_{\text{rel}}$, and relative memory usage under two regimes: (i) fixed rollout budget $G_{\text{total}}=16$; and (ii) a higher-budget variant with $G_{\text{total}}=32$.

Under the fixed-budget regime ($G_{\text{total}}=16$), increasing $k$ monotonically enlarges $\Delta_{\text{div}}$, confirming that larger ensembles provide more diversity. However, Avg@8 and Pass@8 improve only marginally from $k=2$ to $k=4$, and begin to saturate or slightly regress at $k=8$, where each sub-policy receives only $G_k=2$ rollouts and per-model noise offsets the diversity gains. All three configurations have nearly identical FLOPs-level cost, but memory grows roughly linearly with $k$.

When the rollout budget is increased to $G_{\text{total}}=32$, the $k=4$ configuration achieves the best Avg@8 and Pass@8, showing that RLER continues to benefit from more compute in a high-budget regime, at the price of higher VRAM.

*Table 13.* Ensemble-size scaling on DAPO-MATH-17K with QWEN2.5-MATH-7B.

| Setting | $k$ | $G_k$ | Avg@8 | Pass@8 | $\Delta_{\text{div}}$ | $\tilde{C}_{\text{rel}}$ | Memory |
|---|---|---|---|---|---|---|---|
| $G_{\text{total}} = 16$ (fixed budget) | | | | | | | |
| RLER | 2 | 8 | 37.5 | 52.8 | 0.038 | 0.83 | $\sim 2\times$ |
| RLER | 4 | 4 | 37.6 | 53.5 | 0.045 | 0.81 | $\sim 4\times$ |
| RLER | 8 | 2 | 37.2 | 54.0 | 0.051 | 0.80 | $\sim 8\times$ |
| $G_{\text{total}} = 32$ (increased budget) | | | | | | | |
| RLER | 4 | 8 | 37.9 | 53.7 | 0.046 | 0.73 | $\sim 4\times$ |

**Why we choose $k=2$ by default.** Our goal is not only to slightly improve accuracy on a fixed benchmark, but to provide a stable, scalable RLIR alternative for unlabeled, resource-constrained scenarios. From this perspective, $k=2$ is the most practical operating point: it already brings clear gains in accuracy and bias reduction (lower $\rho_{\text{noise}}$, $\rho_{\text{selfbias}}$, $\rho_{\text{symbias}}$) over single-model RLIR, yields smooth scaling curves (Fig. 7), and keeps both FLOPs and memory within a reasonable budget. Larger ensembles offer only marginal improvements under a fixed rollout budget while multiplying VRAM usage, so we recommend $k=2$ as the default choice in practice.

## B.4 Additional Results for Decoupling Experiments

In Section 3, we introduced a set of decoupling experiments where we synthetically controlled the overall noise level $\rho_{\text{noise}}$, the policy–reward coupling $\rho_{\text{selfbias}}$, and the asymmetric drift $\rho_{\text{symbias}}$. Those results showed that: (i) increasing $\rho_{\text{noise}}$ slows convergence and lowers the final accuracy; (ii) policy-dependent noise (self-rewarding) is substantially more harmful than policy-independent noise under the same $\rho_{\text{noise}}$; and (iii) over-reward skew (FP-dominated noise) is much more detrimental than under-reward skew (FN-dominated noise), even when the total noise mass is matched.

Here we provide an additional, more direct study of how false-positive (FP) and false-negative (FN) reward errors affect the final performance of RLIR methods in practice. Concretely, we take Self-Consistency (SC) as the underlying RLIR algorithm on ARITHMETIC dataset, and *correct* its rewards during training using the oracle labels. At each training step,

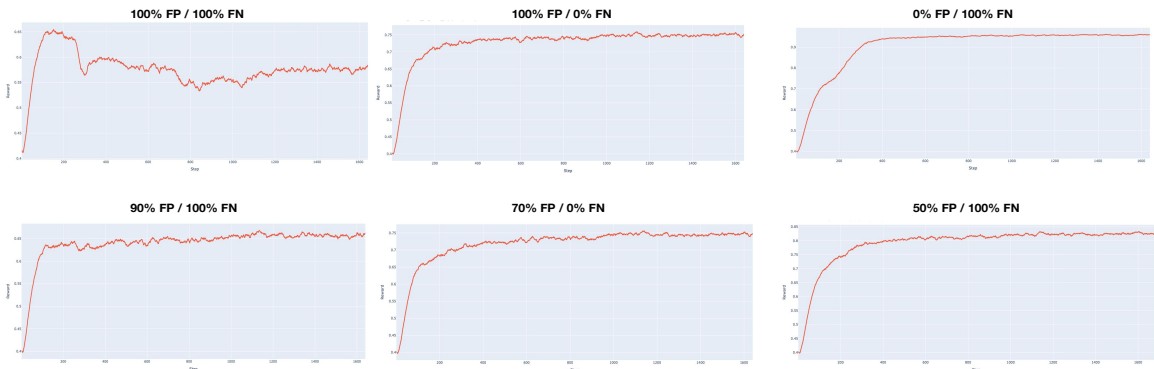

*Figure 9.* We start from the SC baseline and manually correct different fractions of reward labels to the oracle. Each panel is annotated as "$x\%$ FP/$y\%$ FN", indicating the *remaining* proportion of false-positive and false-negative rewards.

after computing the rewards used by SC, we use the oracle to identify FPs and FNs. We then consider the following variants: (1) correcting $0\%$ / $10\%$ / $30\%$ / $50\%$ / $100\%$ of FP rewards (chosen uniformly at random among all FPs) to their correct oracle values, and (2) correcting $100\%$ of FN rewards while leaving all FPs unchanged. Figure 9 report the full learning curves and final test accuracies.

We observe that correcting different fractions of FP rewards yields final test accuracies of $57.6\%$, $66.6\%$, $75.4\%$, $83.1\%$, and $96.0\%$ for $0\%$, $10\%$, $30\%$, $50\%$, and $100\%$ FP correction, respectively. In contrast, correcting *all* FN rewards leads to a final accuracy of $74.8\%$. Thus, even fully eliminating FN errors cannot match the performance obtained by partially correcting FP errors. These results provide concrete empirical evidence that FP-dominated over-reward noise is the primary bottleneck for RLIR, and justify why RLER focuses on suppressing FP bias while also recovering under-rewarded FNs

within its unified reward space.

## C  Proof of Theorem in §3.3

**Setup.**  Given the policy $\pi_\theta$ and the labeling map $\ell : \mathcal{Y} \to \{0, \dots, L-1\}$, define the label probability

$$q_j := \sum_{y_{1:T} : \ell(y_{1:T}) = j} \prod_{t=1}^{T} \pi_\theta(y_t \mid x, y_{<t}), \qquad q_j \geq 0, \qquad \sum_{j=0}^{L-1} q_j = 1.$$

Let the predicted (MAP) label be $m = \arg\max_j q_j$, and write

$$a := q_t, \qquad b := q_m, \qquad o := 1 - a - b.$$

**Hard vs. Soft rewards.**  For a rollout $y_i$ with label $\ell(y_i)$, define

$$r_i^{\mathrm{H}} = \mathbf{1}[\ell(y_i) = m], \qquad \mu_{\mathrm{H}} = b, \quad \sigma_{\mathrm{H}}^2 = b(1-b),$$

$$r_i^{\mathrm{S}} = q_{\ell(y_i)}, \qquad S_2 := \sum_j q_j^2, \quad S_3 := \sum_j q_j^3, \qquad \mu_{\mathrm{S}} = S_2, \quad \sigma_{\mathrm{S}}^2 = S_3 - S_2^2.$$

When the intrinsic probabilities are instantiated by the empirical outcome frequencies $q_j = p_j$, the Soft reward $r_i^{\mathrm{S}} = q_{\ell(y_i)}$ reduces to the Frequency-based method, whereas the Hard reward $r_i^{\mathrm{H}} = \mathbf{1}[\ell(y_i) = m]$ coincides with Self-Consistency.

**Aadvantage and correlation criterion.**  For a group $\{r_i\}_{i=1}^G$, GRPO uses group-wise standardized advantages

$$\bar{r} = \tfrac{1}{G} \sum_i r_i, \qquad s = \sqrt{\tfrac{1}{G} \sum_i (r_i - \bar{r})^2}, \qquad A_i = \frac{r_i - \bar{r}}{s},$$

Because correlation is affine-invariant, replacing population $(\mu, \sigma)$ by group statistics $(\bar{r}, s)$ leaves the comparison unchanged. Hence, with standardized variables,

$$\mathrm{MSE}(r) = \tfrac{1}{G} \sum_i (A_i - A_i^\star)^2 = 2(1 - \rho(r, r^\star)),$$

so that

$$\mathrm{MSE}(r^{\mathrm{S}}) \leq \mathrm{MSE}(r^{\mathrm{H}}) \iff \rho_{\mathrm{S}} \geq \rho_{\mathrm{H}}.$$

When $m \neq t$, both correlations are negative; larger is better.

**Closed forms for $m \neq t$.**  A direct calculation yields

$$\rho_{\mathrm{H}} = \frac{\mathrm{Cov}(r^{\mathrm{H}}, r^\star)}{\sigma_{\mathrm{H}} \sigma_\star} = -\sqrt{\frac{ab}{(1-a)(1-b)}},$$

and, using $\mathbb{E}[r^{\mathrm{S}} r^\star] = a^2$,

$$\mathrm{Cov}(r^{\mathrm{S}}, r^\star) = a^2 - aS_2 = -a(S_2 - a) < 0, \qquad \rho_{\mathrm{S}} = -\frac{a(S_2 - a)}{\sqrt{a(1-a)(S_3 - S_2^2)}}.$$

**Tail dispersion monotonicity.**  Fix $(a, b, o)$ induced by $q$. Let $\mathcal{O} = \mathcal{L} \setminus \{m, t\}$ and $s_{\max} = \max_{j \in \mathcal{O}} q_j$. Making the non-majority (tail) mass $o$ more dispersed strictly decreases $S_2 = \sum_j q_j^2$ by convexity of $x^2$ and strictly increases $S_3 - S_2^2$. Therefore $|\rho_{\mathrm{S}}|$ strictly decreases, while $\rho_{\mathrm{H}}$ is unaffected. Hence the *worst case* for $\rho_{\mathrm{S}}$ at fixed $(a, b, o)$ occurs when the tail is fully concentrated, i.e. $s_{\max} = o$.

**Sufficiency.** In the worst case $s_{\max} = o$,

$$\rho_{\text{S}} - \rho_{\text{H}} = (a - s_{\max}) \frac{(1 - b) \sqrt{a(1 - a)}}{\sqrt{ab} \sqrt{S_3 - S_2^2}} > 0 \quad \text{whenever } a \geq s_{\max}.$$

Since tail dispersion only improves $\rho_{\text{S}}$, we have $\rho_{\text{S}} \geq \rho_{\text{H}}$ for all tail configurations whenever $a \geq s_{\max}$.

**Necessity.** If $a < s_{\max}$, concentrate the entire tail mass on a single label so that $s_{\max} = o$. The same expression becomes negative, implying $\rho_{\text{S}} < \rho_{\text{H}}$, i.e. $\text{MSE}(r^{\text{S}}) > \text{MSE}(r^{\text{H}})$.

**Conclusion.** Under $m \neq t$, the Soft reward is closer to the oracle than the Hard reward *if and only if* $a \geq s_{\max}$.

## D Prompt Template for RLER

```
system_prompt

system_prompt: |
  You are a mathematical reasoning expert. When given a math problem, analyze it
    step by step. First, detail your internal reasoning in a <think> block using
    steps (e.g., "Step 1:", "Step 2:", etc.). Then, provide only the final
    conclusion in an <answer> block. Follow this exact format with no extra text:

  <think>
  Step 1: ...
  Step 2: ...
  ...
  </think>
  <answer>
  ...
  </answer>
```

