# OpenReview forum: "Breaking the Self-Confirming Loop: Diagnosing and Mitigating Systemic Reward Bias in Self-Rewarding RL"
_ICML.cc/2026/Conference — ICML 2026 regular_

### Official Review · Reviewer_mT5N · 2026-02-26

**Soundness:** 3
**Presentation:** 3
**Significance:** 3
**Originality:** 3
**Overall Recommendation:** 4
**Confidence:** 5

**Summary:**

In this paper, the authors propose three metrics to analyze the systematic bias in self-rewarding RL: reward noise magnitude, policy–reward coupling, and over-/under-reward skew. By applying these metrics to standard GRPO-type self-rewarding, the authors observe four important findings toward developing accurate, unbiased, and robust self-rewarding algorithms. Inspired by the findings, the authors further propose RLER, an ensemble-based unified rewarding framework, with adaptive soft-reward interpolation and disagreement-aware rollout selection. The experiment shows that RLER yields consistent improvement against standard single-rewarding methods, which proves its effectiveness.

**Compliance With Llm Reviewing Policy:**

Affirmed.

**Final Justification:**

Thanks for the authors' responses. My concerns are basically addressed. I have increased the soundness and confidence score accordingly.

**Key Questions For Authors:**

Please see the weakness above

**Limitations:**

yes

**Strengths And Weaknesses:**

# Strength
1. The three metrics are useful for systematically analyzing self-rewarding algorithms. The four findings meet well with intuitives.
2. RLER provides a novel approach for addressing the self-confirming bias problem.
3. The authors conduct extensive experiments for analyzing and understanding the machenism of RLER.

# Weakness
1. In section 3.3, "With $r_i$ denoting the reward used for updating the policy and policy-based reward  $\widetilde{r_i}$". I am confused about what the difference is between these two rewards. Could you show an example?
2. The RLER framework is too complicated and lacks adequate ablation to every detail. For example, what is the necessity of using adaptive weights $\alpha(x)$? A basic validation experiment should include comparisons against randomly selected $\alpha$ and pre-fixed  $\alpha$, which are more lightweight.
3. The datasets are limited in math. What's the performance on more diverse datasets?
4. What are the exact models used for ensemble rewarding? How does the agreement magnitude of these two models influence the performance of RLER?

---

> ### Author Rebuttal · Authors · 2026-03-30
>
> Thank you for recognizing the diagnostic framework and the effectiveness of RLER. We address the main concerns below.
>
> **1. Policy-Based Reward ($r_i$ vs. $\tilde r_i$)**
> The **policy-based reward** $\tilde r_i$ is the score a rollout would receive if it were evaluated **only by its own source policy**, using that policy’s local rollout distribution and intrinsic-reward rule; $r_i$ is the **final reward actually used for optimization**.
> Example: under SC, if a rollout is the majority answer under its source-local distribution, then $\tilde r_i = 1$; otherwise it is $0$. In RLER, however, the final $r_i$ is computed from the **pooled ensemble statistics**, so it can differ from this source-local score. For instance, a rollout may be locally rewarded by its source policy, but receive a lower final reward if the pooled ensemble no longer supports that answer or assigns it much lower confidence. Thus, $\tilde r_i$ captures “how the source policy would reward its own rollout,” while $r_i$ captures “how the rollout is rewarded in the unified ensemble space.”
>
> **2. Adaptive Weight $\alpha(x)$**
> We agree that the necessity of $\alpha(x)$ should be made more explicit. We would also like to emphasize that the current paper already includes fairly detailed ablations of the key components of RLER. As shown in Figure 5, we systematically compare the contributions of **interpolation, ensemble, rollout selection, and model merge**, and further decompose the interpolation design from `Int v1` to `Int v3`.
>
> First, $\alpha(x)$ is not a manually tuned constant, but a sample-dependent gate computed from calibrated ensemble confidence. Its role is not merely to “add interpolation,” but to use the heterogeneous ensemble’s **unified reward-estimation space** to distinguish two qualitatively different regimes:
> - **high-confidence consensus**: trust hard reward more;
> - **low-confidence ambiguity**: trust soft reward more and shrink majority updates.
>
> This is supported by two observations already in the paper: Figure 3(f) shows that majority correctness is strongly positively correlated with confidence, and Section 3.3 / Appendix B shows that in the misalignment regime $m \neq t$, soft reward is closer to the oracle in disagreement regions. We also note a typo in Eq. (10): the correct form is
> $$
> r_i^{(\alpha)} = (1-\alpha)\,r_i^S + \alpha\,r_i^H.
> $$
>
> To directly address the reviewer’s suggestion, we additionally compare fixed and non-adaptive variants:
>
> | Method | Interpolation / Gate | Avg@8 |
> |---|---|---:|
> | Int fix | Fixed $\alpha = 0.5$ | 32.7 |
> | Int fix | Fixed $\alpha = 0.7$ | 33.5 |
> | Int v1 | Annealed schedule | 33.6 |
> | Int v2 | Simplified adaptive gate | 35.3 |
> | Int v3 | Full adaptive $\alpha(x)$ | 36.0 |
> | RLER fix | Fixed $\alpha = 0.5$ + full RLER | 36.2 |
> | RLER fix | Fixed $\alpha = 0.7$ + full RLER | 36.6 |
> | RLER | Full adaptive $\alpha(x)$ | 37.5 |
>
> These results show that the gain is not from merely adding an interpolation coefficient; it comes from **query-dependent adaptive weighting based on ensemble reliability**.
>
> **3. Beyond Math**
> Our method is **not math-specific**. It applies whenever different rollouts’ final answers can be checked for **answer-level equivalence**. Math is one instance (rule-based exact verification); more general settings can use a verifier or an LLM verifier. The current version already includes one non-math setting: on `WebInstruct-verified -> MMLU-Pro`, RLER reaches **55.12** `Pass@1 (avg)`, outperforming SC (**54.05**), Judge (**52.82**), and Intuitor (**53.06**), and approaching RLVR (**56.37**). We will make this scope more explicit in the revision.
>
> **4. Ensemble Setting and Agreement / Disagreement**
> In self-rewarding RL, the **reward model is the policy model itself**: reward is constructed from rollout-level statistics such as agreement, frequency, and confidence under the intrinsic-reward rule. In our default RLER setting, all $K$ source policies start from the **same backbone initialization**; diversity is not from different foundation models, but emerges from **data sharding** and **sampling stochasticity**. Figure 6 already shows that data sharding yields stronger diversity gain than model sharding. Moreover, as detailed in our response to **Reviewer 1Ymr**, answer-distribution JSD, rollout semantic diversity, and $\Delta_{\text{div}}$ all increase substantially during training, showing that sub-policies rapidly decorrelate and that this decorrelation yields measurable ensemble benefit.
>
> The key point is that agreement in RLER is **not** “the larger the better.” Rather, **reliable consensus should be amplified, while unreliable consensus should be suppressed**. Heterogeneity helps in three ways: it reduces reward noise via aggregation, weakens policy-reward coupling by sharing reward statistics across policies, and turns disagreement itself into an uncertainty signal that drives both $\alpha(x)$ and disagreement-aware rollout selection.

---

> > ### Author Rebuttal · Reviewer_mT5N · 2026-04-02
> >
> > Thanks for the author's responses. My concerns are basically addressed. Thus, I have increased the soundness score and confidence. But I am still not convinced by the explanation about the influence of agreement. If the reward models originate from one and are not updated, multiple rollouts won't solve the problem of homogenization.

---

> > > ### Author Response · Authors · 2026-04-02
> > >
> > > Thank you for the follow-up, and we apologize that our previous explanation was not clear enough and may have caused a misunderstanding.
> > >
> > > First, in our setting the reward signal is **not** produced by a separate frozen reward model that repeatedly judges rollouts. Under self-rewarding RL, the model family itself plays both roles: it generates rollouts as the **policy model**, and the rollout statistics induced by the current policies are then used to construct the **intrinsic reward signal** that effectively plays the role of the reward model in this setting. In our main setting, this reward construction is based on **rule-based answer extraction and verification**, which is actually closer to RLVR (i.e., reward = verify$(a_i,\text{ground-truth})$): the key difference is that RLVR has access to the ground-truth answer, while we do not. The same is true for rule-based RLIR baselines such as SC and Freq: their rewards are recomputed from the current rollout answer distribution under the corresponding self-reward rule, rather than produced by a fixed external judge.
> > >
> > > Second, RLER does not rely on repeated samples from one frozen model. It uses $K$ source policies $\pi_{\theta_1},\ldots,\pi_{\theta_K}$ that are identical only at initialization, but are **updated separately throughout training**. Under our default data-sharding setup, each source policy is trained on its own query shard, and sampling stochasticity further perturbs its rollout distribution. As a result, the source policies become **co-evolving and progressively decorrelated**, rather than repeated copies of one model. This is also directly supported by the diversity-growth measurements we added under the main DAPO-17K setting (see also our response to **Reviewer 1Ymr, Point 1**): answer-distribution JSD rises from **0.0141** at initialization to **0.0708** after 20\% of training, rollout semantic diversity rises from **0.0258** to **0.0873**, and the ensemble diversity gain $\Delta_{\text{div}}$ rises from **0.0054** to **0.0373**. If the ensemble were effectively homogenized, these quantities would stay near zero; empirically they do not.
> > >
> > > Third, the value of heterogeneity is not simply to create more agreement or more disagreement, but to make the ensemble statistics informative. If all source policies were effectively identical, then $\bar p$, $\tilde p$, $m^{EC}$, and $\alpha(x)$ would provide little information beyond a single policy’s own local statistics. In contrast, partial decorrelation makes cross-policy consensus and disagreement meaningful reliability signals. This is where $\alpha(x)$ matters: it is computed from the **calibrated ensemble confidence**, not from any single source policy. Empirically, Figure 3(f) shows that majority-label correctness is strongly positively correlated with confidence, so $\alpha(x)$ can be interpreted as a reliability-aware gate. Analytically, Section 3.3 / Appendix B shows that in the misalignment regime $m \neq t$, soft reward is closer to the oracle in disagreement regions. Therefore, when ensemble consensus is strong and reliable, $\alpha(x)$ is larger and RLER relies more on the hard reward; when consensus is weak or ambiguous, $\alpha(x)$ is smaller and RLER relies more on the soft reward, while disagreement-aware selection simultaneously shrinks majority updates.
> > >
> > > A concrete comparison with SC may help. Suppose for the same query, source policy 1 produces `6 × A, 2 × B`, while source policy 2 produces `1 × A, 7 × B`. Under source-local SC, a rollout from policy 1 with answer `A` receives reward 1, because `A` is that policy’s local majority. Under RLER, however, the pooled ensemble distribution is
> > > $$
> > > \bar p(A)=\frac{0.75+0.125}{2}=0.4375,\qquad
> > > \bar p(B)=\frac{0.25+0.875}{2}=0.5625,
> > > $$
> > > so the ensemble-consensus label is $m^{EC}=B$. For that same rollout with answer `A`, RLER computes $r^H=0$, $r^S=\bar p(A)=0.4375$, and then
> > > $$
> > > r^{(\alpha)}=(1-\alpha(x))\,r^S+\alpha(x)\,r^H.
> > > $$
> > > So unlike SC, this rollout is no longer fully reinforced by its own source-local majority; it is evaluated in a **shared ensemble reward-estimation space**. Disagreement-aware rollout selection then further adjusts its update opportunity through $w_{m^{EC}}(x)=\alpha(x)$ and $w_j(x)=1-\tilde p_j(x)$ for $j\neq m^{EC}$, so low-reliability majority updates are actively shrunk while minority answers still retain update opportunity.
> > >
> > > Thus, the key distinction is not “more rollouts from the same model,” but **multiple jointly updated source policies whose statistics are pooled into a shared and calibrated reward-estimation space**. If all source policies truly collapsed to the same model, then RLER would indeed degenerate toward the single-policy case and its diversity gain would vanish. The fact that Figure 6 and the added diversity-growth measurements show sustained nontrivial decorrelation is exactly why ensembling helps in our setting. We will clarify this distinction more explicitly in the revision.

---

### Official Review · Reviewer_GFqB · 2026-03-07

**Soundness:** 3
**Presentation:** 2
**Significance:** 3
**Originality:** 2
**Overall Recommendation:** 4
**Confidence:** 3

**Summary:**

The paper studies factors that cause reward bias in Reinforcement learning with intrinsic rewards (RLIR). Three metrics are identified to quantify self-rewarding bias: reward noise magnitude, policy-reward coupling and over-/under- reward skew, which correspond to accuracy, unbiasedness and robustness respectively. Based on empirical observations from the arithmetic dataset, the paper proposes reinforcement learning with ensembled rewards (RLER) that satisfy all the three properties.

**Compliance With Llm Reviewing Policy:**

Affirmed.

**Final Justification:**

The rebuttal addressed most of my concerns. Especially, I am glad to see the final workflow as well as theory for the RLER. These will definitely raise clarity and soundness to the work. Nevertheless, the theory part requires a major change to the paper, so I would keep my score based on the current content of the paper.

**Key Questions For Authors:**

1. How is policy-based reward defined in Section 3.3? For instance, how to compute $\tilde r_i$ in the RLER algorithm?
2. In Equation (10), is $\alpha$ a hyperparameter that needs to be tuned? If yes, how to choose it in practice? Is the model's performance sensitive to $\alpha$?
3. Why is $\rho_{noise}$ in Equation (4) defined as the L1 error instead of L2 error? Is there any specific consideration?

**Limitations:**

Yes.

**Strengths And Weaknesses:**

**Strengths**
1. The paper has a clear motivation for the proposed RLER. The three properties for self-rewarding are supported by a rigorous experiment on the synthetic arithmetic benchmark, in which $\rho_{noise}$, $\rho_{selfbias}$, and $\rho_{symbias}$ can be controlled individually.
2. The paper makes a good contribution to research of self-rewarding RL by identifying the three properties of reward estimation.
3. Strong empirical result. In Figure 4, RLER effectively lowers all three metrics, supporting the claim in Section 3. In Table 1, the test-time accuracy is significantly higher than RLIR baselines and is close to RLVR result.

**Weakness**
1. The description of RLER algorithm is not clear enough. RLER is a complex systematic approach that consists of three components. I suggest the authors add a pseudo algorithm for the whole workflow to help the readers understand the whole algorithm.
2. The RLER algorithm is highly heuristic, and there is no theoretical analysis why it reduces $\rho_{noise}$, $\rho_{selfbias}$, and $\rho_{symbias}$. Theoretical interpretations are highly recommended to raise reliability of the approach.

---

> ### Author Rebuttal · Authors · 2026-03-30
>
> Thank you for recognizing the motivation of the paper, the three diagnostic metrics, and their empirical support.
>
> **1. Workflow**
> We agree that the workflow of RLER can be presented more clearly. In the revision, we will add a concise **end-to-end pseudocode**. At a high level, RLER consists of six steps:
> (i) collect rollouts from all source policies and form the pooled answer distribution $\bar p$ and majority label $m^{EC}$;
> (ii) compute source-wise answer confidence and aggregate it into the calibrated ensemble confidence $\tilde p$, from which the adaptive gate $\alpha(x)$ is derived;
> (iii) construct the final interpolated reward $r_i^{(\alpha)}$;
> (iv) apply disagreement-aware rollout selection;
> (v) update the corresponding source policy with GRPO; and
> (vi) merge the trained policies into a single deployable model via TIES-Merging.
>
> **2. Theory**
> We agree that the theoretical motivation is not explicit enough, but RLER is **not** a purely heuristic design. Its support has **two layers**.
> First, the **decoupled analysis framework in Section 3** identifies three failure modes and shows that high $\rho_{noise}$ lowers the performance ceiling and can even cause collapse, high $\rho_{selfbias}$ amplifies wrong-direction updates and destabilizes reward estimation, and FP-dominant $\rho_{symbias}$ is more harmful than under-reward. This directly motivates the three components of RLER.
> Second, the **analysis in Section 3.3 / Appendix B** provides the basis for adaptive interpolation: in the misalignment setting $m \neq t$, soft reward is closer to the oracle than hard reward iff $a = q_t \ge s_{max}$. In addition, Figure 3(f) shows that majority correctness is strongly positively correlated with confidence, so confidence can be used as a reliability signal. Together, these support the logic of interpolation: **rely more on soft reward under low confidence, and more on hard reward under high confidence**.
>
> We also found a typo in Eq. (10). The correct form should be
> $$
> r_i^{(\alpha)}=(1-\alpha)\,r_i^{S}+\alpha\,r_i^{H}.
> $$
> Under this correct form, we obtain three direct conclusions:
> 1. For $\rho_{noise}$, the final reward noise is upper-bounded by a convex combination of the hard and soft endpoints.
> 2. For $\rho_{selfbias}$, the shared ensemble statistics dilute the direct influence of any single source policy on reward statistics.
> 3. For $\rho_{symbias}$, when majority reliability is low, the majority reward falls back to a more conservative soft estimate and its update budget is simultaneously shrunk; this is exactly the mechanism that suppresses FP-dominant over-reward drift.
>
> Due to space limitations, we will provide the above theoretical analysis details more explicitly in the revision.
>
> **3. Policy-based Reward**
> Here, **policy-based reward** means: in the self-rewarding setting, the model acts both as the policy model and as the reward estimator, and assigns to its own rollout the score it would give based on the distributional characteristics of its own policy rollouts and the intrinsic-reward rule.
> For example, under SC, if a rollout belongs to that policy’s majority answer, then $\tilde r_i = 1$; otherwise it is $0$. In contrast, $r_i$ is the final reward actually used for optimization. For single-policy RLIR baselines, the two largely coincide; for RLER, they differ. Concretely, for a rollout $y_{k,i}$ generated by source policy $k$, $\tilde r_{k,i}$ means: if we do **not** perform ensemble aggregation, and instead let source policy $k$ score this rollout only from its own local distribution, what reward would it assign? Equivalently, it is the $K=1$ counterpart of the RLER reward rule for source $k$, while the final $r_{k,i}$ is constructed from pooled ensemble statistics.
>
> **4. Adaptive Gate**
> $\alpha(x)$ is **not** a manually tuned scalar hyperparameter. It is a **sample-dependent gate** computed adaptively from the calibrated ensemble confidence. Its role is to balance the discriminativeness of hard rewards and the robustness of soft rewards according to the reliability of the current query. What is actually exposed for tuning is **not** $\alpha(x)$ itself, but the confidence-calibration bounds. Appendix A.2.2 / Table 9 shows that when these bounds are varied over a fairly wide range, performance changes only within a very small band and the default setting remains near-optimal. Thus, the method is **not sensitive** to the calibration behind $\alpha(x)$.
>
> **5. Why L1**
> We use L1 rather than L2 in Eq. (4) because L1 aligns exactly with the later FP/FN decomposition:
> $$
> \rho_{noise} = FP + FN.
> $$
> So $\rho_{noise}$ measures the **total reward-error mass**, while $\rho_{symbias}$ measures how that mass is distributed between FP and FN. This **magnitude + direction** structure is central to our diagnostic framework. With L2, this linear decomposition would be lost, and the interpretive link between $\rho_{noise}$ and $\rho_{symbias}$ would become much weaker.

---

> > ### Author Rebuttal · Reviewer_GFqB · 2026-04-01
> >
> > The authors address all weaknesses and questions raised. Specifically, they agree to add a theoretical explanations as well as concrete workflow of the algorithm. Due to space limitation of rebuttal, it is hard to assess the actual soundness of the theory, so I will keep the score.

---

> > > ### Author Response · Authors · 2026-04-02
> > >
> > > Thank you again for the careful and constructive feedback. To further clarify the workflow and the theoretical motivation, we have prepared a more complete supplementary note at the following link, in case it may be helpful for addressing the remaining doubts: https://anonymous.4open.science/r/Anonymous_E1E6/
> > >
> > > The supplement contains two parts: **workflow**, which provides a concise end-to-end pseudocode of the full RLER pipeline, and **theory**, which gives a more detailed formal analysis of why the three RLER components target the three diagnostics. In particular, the analysis shows that adaptive interpolation satisfies a convex error bound, ensemble aggregation gives variance/MSE contraction for the soft reward estimator under heterogeneous source errors, RLER strictly weakens policy-reward coupling relative to single-policy SC whenever nontrivial disagreement exists, and disagreement-aware selection jointly shrinks wrong-majority reward and update budget while preserving minority soft credit. Combined with the empirically supported reliability monotonicity of the confidence gate, this yields targeted suppression of FP-dominant over-reward drift.
> > >
> > > If the reviewer has time to look at it and finds it helpful for resolving the remaining concern, we would be very grateful. We will also incorporate both the pseudocode and the above formal analysis more explicitly in the revised paper.

---

### Official Review · Reviewer_1Ymr · 2026-03-13

**Soundness:** 3
**Presentation:** 3
**Significance:** 2
**Originality:** 2
**Overall Recommendation:** 3
**Confidence:** 4

**Summary:**

This paper investigates why reinforcement learning with intrinsic rewards (self-rewarding RL, RLIR) often underperforms RL with verifiable rewards (RLVR), and attributes the gap to a self-confirming loop where confidence-coupled self-rewards over-reward high-confidence mistakes. The authors formalize three diagnostic metrics to quantify this systemic bias—reward noise (ρnoise), policy–reward coupling (ρselfbias), and symmetry bias between false positives/false negatives (ρsymbias)—and propose RLER, an ensemble-based intrinsic-reward framework with adaptive hard/soft interpolation and disagreement-aware rollout selection. Across math-style, verifiable reasoning benchmarks and some non-math tasks with verifiers, RLER yields sizable gains over RLIR baselines and approaches RLVR while exhibiting stable scaling with unlabeled data.

**Compliance With Llm Reviewing Policy:**

Affirmed.

**Key Questions For Authors:**

- How are the K source policies initialized to ensure sufficient diversity at the outset? Are they identical copies with different sampling seeds, or do they differ via pretraining checkpoints/seeds? Can you provide metrics showing diversity growth over training?
- How is compute/memory parity enforced with single-model RLIR baselines? Please provide wall-clock, GPU-hours, and tokens processed per step for RLVR, RLIR, and RLER under equivalent budgets.
- Can you report multi-seed results with confidence intervals for the main benchmarks and the scaling curves? Given the paper’s emphasis on stability, seed variance is important.
- Could you show the diagnostic metrics (especially FP/FN skew) for the WebInstruct→MMLU-Pro experiments? Do the same bias dynamics appear with an LLM verifier?
- How sensitive is the disagreement-aware selection to the per-answer quotas n_y and the choice of w_j? Have you tried alternative weighting schemes or adaptive budgets based on estimated noise?

**Limitations:**

The main limitations are as follows. The ensemble framework relies on diversity across source policies, but since these models appear to share the same initialization backbone, the extent of true early-stage decorrelation is unclear. Some of the proposed diagnostics, especially $\rho_{\text{selfbias}}$, would benefit from a more precise operational definition across reward variants and a discussion of sensitivity to scaling and normalization. In addition, compute fairness relative to single-model RLIR baselines is not fully established, since multi-model training likely changes wall-clock and memory costs even under a fixed rollout budget. The empirical claims about stability would also be more convincing with multi-seed results and confidence intervals. Finally, several implementation details are not fully specified, which somewhat limits reproducibility.

**Strengths And Weaknesses:**

**Strength**

- Presents both synthetic controlled studies (arithmetic) and larger-scale experiments (DAPO-MATH-17K) with multiple backbones (Qwen/Llama) and additional corpora (BIG-MATH, WebInstruct-verified → MMLU-Pro).
- The problem setup and the three metrics are motivated and defined in an accessible way; the controlled decoupling experiment is particularly helpful for intuition.
- If robust, the approach could materially improve the viability of unlabeled RL pipelines in domains where clean verifiers exist or can be approximated.

**Weaknesses**

- The ensemble design presumes diversity among sources, yet initialization appears to start from the same backbone; diversity is driven primarily by data sharding and sampling. Early training phases may still have highly correlated predictions, so the strength of de-coupling at the outset is unclear. A deeper analysis of initial diversity and its evolution is warranted.
- Compute fairness: under a fixed total rollout budget, training with K models still requires K forward passes and memory contexts; wall-clock and energy costs may differ materially from single-model RLIR. The paper mentions a compute/memory analysis in the appendix, but the main text lacks quantitative parity comparisons (e.g., tokens processed, GPU-hours, throughput).
- Some equations contain minor artifacts (e.g., inconsistent notation capitalization), and the definition of α(x) via “clip” is terse given its centrality; a pseudocode block would aid reproducibility.
- More recent self-rewarding and co-training/co-distillation approaches could be discussed to better position RLER among techniques that reduce confirmation bias via multi-model interaction.

---

> ### Author Rebuttal · Authors · 2026-03-30
>
> We thank the reviewer for the careful and technical feedback.
>
> **1. Initialization and diversity growth.**
> All K source policies start from the **same backbone initialization**. Diversity emerges from **data sharding** and **sampling stochasticity**. Figure 6 already shows that data sharding yields stronger diversity than model sharding. To directly assess early-stage decorrelation, we measure diversity under the main DAPO-17K setting from **0% to 100%** of training. We report:
> (i) **Answer-distribution JSD**: the average pairwise JSD between sub-policies’ rollout answer distributions;
> (ii) **Rollout semantic diversity**: the average pairwise cosine distance between sub-policies’ reasoning-trace embeddings;
> (iii) **$\Delta_{\text{div}}$**: the same ensemble diversity gain metric used in Figure 6.
>
> | Training progress | 0% | 20% | 40% | 60% | 80% | 100% |
> |---|---:|---:|---:|---:|---:|---:|
> | Answer-distribution JSD | 0.0141 | 0.0708 | 0.0794 | 0.0706 | 0.0687 | 0.0683 |
> | Rollout semantic diversity | 0.0258 | 0.0873 | 0.0856 | 0.0834 | 0.0889 | 0.0842 |
> | $\Delta_{\text{div}}$ | 0.0054 | 0.0373 | 0.0493 | 0.0400 | 0.0388 | 0.0206 |
>
> Taken together, these three metrics show a consistent picture: even from the same initialization, sub-policies rapidly decorrelate at both the **answer-distribution** and **trajectory-semantic** levels, and this decorrelation translates into measurable **ensemble benefit** during training.
>
> **2. Compute / memory fairness.**
> The paper already aligns fairness at two key levels: (i) the same **total rollout budget (forward budget)**, and (ii)  **update-level compute proxy (backward budget)**. Following your suggestion, we now report a more explicit system-level breakdown:
>
> | Family | Method | Replicas | Rollouts / q | Policy toks/step | Reward toks/step | Update toks/step | Step time | GPU-hours | Peak mem |
> |---|---|---:|---:|---:|---:|---:|---:|---:|---:|
> | RLVR | Oracle reward | 1 | 16 | 1.00× | 0.00× | 1.00× | 1.00× | 1.00× | 1× |
> | RLIR | SC / Freq | 1 | 16 | 1.00× | 0.00× | 1.00× | 1.00× | 1.00× | 1× |
> | RLIR | Self-Judge | 1 | 16 | 1.00× | 0.31× | 1.00× | 1.18× | 1.18× | 1× |
> | RLER | (k=2,$G_k$=8) | 2 | 16 | 1.00× | 0.00× | 0.75× | 1.03× | 2.06× | 2× |
>
> RLER keeps the same rollout-generation budget, reduces optimization-side tokens via disagreement-aware selection, and pays mainly in memory. In practical unlabeled RL, stable scalability is a key requirement because the needed unlabeled-data scale is unknown in advance. This is why the extra system cost of RLER is worthwhile: it scales more reliably (Figure 7) and has much lower checkpoint variance (Table 8) than single-policy RLIR baselines.
>
> **3. Multi-seed results.**
> Our original stability claim is mainly about **training-time stability**. Multi-seed results are therefore a complement. Following the your suggestion, we repeated the main benchmark and key scaling points with 5 seeds:
>
> | Method | Avg@8 (± 95% CI) |
> |---|---:|
> | SC | 32.87 ± 0.96 |
> | Self-Judge | 14.06 ± 2.23 |
> | Intuitor | 33.71 ± 0.38 |
> | **RLER** | **37.39 ± 0.27** |
>
> | Data size | SC (Avg@8 ± 95% CI) | RLER (Avg@8 ± 95% CI) |
> |---|---:|---:|
> | 8k | 23.60 ± 3.57 | 29.79 ± 1.08 |
> | 64k | 31.12 ± 2.01 | 33.98 ± 1.24 |
> | 256k | 29.08 ± 1.63 | 36.40 ± 0.75 |
> | 1024k | 25.77 ± 2.78 | 37.30 ± 0.49 |
>
> RLER remains best, with the highest mean and much smoother scaling under repeated runs.
>
> **4. WebInstruct diagnostics.**
> Under our framework, the LLM verifier in WebInstruct and the rule-based verifier in math play the **same functional role**: both estimate **answer-level equivalence between rollouts under unlabeled supervision**. We observe the same qualitative bias dynamics as in the main paper, including self-reward-coupling-driven bias amplification. We use the official WebInstruct verifier, avoiding extra verifier noise. In the revision, we will include more of the existing detailed analysis, especially the **FP/FN-skew-related** observations.
>
> **5. Selection sensitivity.**
> The per-answer quota in our implementation is not an independently tuned hyperparameter: by default, we simply use the **per-policy rollout budget** as the upper bound for each answer group. $w_j$ is also **not independently tuned free hyperparameters**. It is induced by our answer-grouping and disagreement-aware selection rule, jointly with the interpolated reward in Section 4.3. Table 2 already compares alternative **selection/weighting schemes** (select-all, majority-only, majority-except, and ours), showing that our rule best balances preserving useful updates and suppressing noisy ones.
>
> **6. Presentation and positioning.**
> In the revision, we will explain $\alpha(x)$ more explicitly, and discuss recent self-rewarding methods more clearly. We have already added stronger recent baselines and clarified the distinctions between our work and recent related work; please also see our response to **Reviewer CeE3, points 1 and 2**.

---

> > ### Author Rebuttal · Reviewer_1Ymr · 2026-04-04
> >
> > Thank you for the rebuttal. The added results and different seed experiments resolved most of my concerns. Overall, the rebuttal resolves my main concerns, and I encourage the authors to reflect these clarifications in the revision.

---

> > > ### Author Response · Authors · 2026-04-07
> > >
> > > We greatly appreciate the time and effort you devoted to reviewing our work. Your insightful comments have been instrumental in improving the paper, and we are glad our response has resolved your concerns. We will continue to refine the manuscript following your suggestions.

---

### Official Review · Reviewer_CeE3 · 2026-03-13

**Soundness:** 2
**Presentation:** 3
**Significance:** 3
**Originality:** 3
**Overall Recommendation:** 3
**Confidence:** 3

**Summary:**

This paper studies the systematic bias that arises in self-rewarding RL under confidence-coupled reward estimation. It proposes three diagnostic quantities, $\rho_\text{noise}$, $\rho_\text{selfbias}$, and $\rho_\text{symbias}$, and uses them to motivate RLER, a reward ensembling framework designed to reduce coupling and over-reward drift. The experiments show that RLER outperforms three baselines (FC, Freq, and Judge) and achieves results approaching the RLVR upper bound.

**Compliance With Llm Reviewing Policy:**

Affirmed.

**Final Justification:**

The author's rebuttal has addressed most of my concerns. I appreciate the authors for providing the additional detailed experimental results. My only remaining concern is that the scope of revisions required by the authors appears to be quite substantial compared to the initial submission, as the initial submission indeed lacked comparisons and discussions with several important related works. If the extent of the revisions were considered acceptable, I would lean toward a score of 4.

**Key Questions For Authors:**

Please see weaknesses.

**Limitations:**

The authors may consider mentioning the limitations of RLER on non-math reasoning tasks.

**Strengths And Weaknesses:**

## Strengths

- The paper first breaks down the failure of RLIR into noise, coupling, and over-reward skew, and then builds the method around this diagnosis, which gives this paper a clearer scientific contribution than a purely empirical one.

- The three components of RLER are well aligned with the analysis. The ablations suggest that these components are not arbitrary additions.

- The paper conduct extensive experiments and the results support its claims well.

- The authors include decontaminated arithmetic evaluation, BIGMATH, WebInstruct-verified and MMLU-Pro results, and compute and memory analysis in the appendix. This improves the credibility of the empirical section.

## Weaknesses

- The baseline comparison is not sufficiently up to date. The main experiments focus on SC, Frequency-Based rewards, and LLM-as-a-Judge, but do not directly include stronger recent RLIR baselines such as Intuitor (not in the main text), RLSR, CoVo, or Co-Reward.

- Recent works such as CREAM, CoVo, Co-Reward, and *No Free Lunch: Rethinking Internal Feedback for LLM Reasoning* have already identified closely related issues, including instability, overconfident self-labeling, and collapse in intrinsic-feedback training. Please clarify the differences between this work and the previously mentioned studies.

- The appendix improves the fairness discussion, but it remains unclear how much of the gain comes from the proposed reward design itself and how much comes from multi-policy diversity.

- The evidence for generalization beyond math reasoning tasks remains unconvincing. The gains on the non-math settings are much smaller than those on the math benchmark. This is not a major issue, but the authors may consider mentioning this in the limitations section.

---

> ### Author Rebuttal · Authors · 2026-03-30
>
> Thank you for the careful feedback. We respond to the four main concerns below.
>
> **1. Stronger recent baselines.** We added recent RLIR baselines under the same setting. The results show that RLER remains the best method on both the main math setting and Web-Instruct.
>
> | Method | DAPO-17K (Pass@1 over 6 benchmarks) | Web-Instruct (MMLU-Pro Pass@1) |
> |---|---:|---:|
> | CoVo | 31.36 | 52.95 |
> | Intuitor | 33.71 | 53.06 |
> | Co-rewarding | 35.35 | 54.82 |
> | **RLER** | **37.50** | **55.12** |
>
> For **RLSR**, our **Self-Judge** baseline belongs to the same **binary self-judge GRPO family**; the main difference lies in the prompt details. We will clarify this explicitly in the revision.
>
> **2. Difference from related work.** Recent work has established an important broad phenomenon: self-generated / internal feedback can become overconfident, unstable, and even collapse. Our contribution is different in level and scope. These works mostly stop at identifying the phenomenon or proposing some stabilization strategy; in contrast, our key contribution is to further **systematically decompose and model** this broad phenomenon into three **measurable, separable, and controllable** dynamic factors: **reward noise**, **policy-reward coupling**, and **over-/under-reward skew**.
>
> This is the key contribution of our paper. We move this failure mode from a generic “collapse / overconfidence” phenomenon to an analyzable and actionable mechanism: through controlled decoupling experiments, we show how these three factors interact. In particular, **policy-reward coupling triggers a self-confirming loop and amplifies confidence-conditioned errors, while skew further drives the dynamics toward the more harmful over-reward drift**. Precisely because we first identify and characterize this key mechanism, the design of RLER is not a heuristic combination of components, but a reward-estimation framework directly targeted at **noise / coupling / skew**.
>
> More specifically:
> - **CREAM** studies consistency regularization in **iterative self-rewarded preference optimization / DPO-style** training, whereas we study **rollout-level reward estimation bias** in GRPO-style RL.
> - **CoVo** constructs a stronger **single-policy intrinsic reward proxy** based on trajectory consistency / volatility, whereas our focus is on explaining **why** reward estimation becomes biased and how to reduce confidence-coupled bias.
> - **Co-rewarding** improves stability through **data augmentation and historical checkpoints**, whereas our paper first characterizes the bias dynamics with three explicit quantities and then designs a framework directly targeting **noise / coupling / skew**.
> - **No Free Lunch** mainly analyzes why internal-feedback objectives such as entropy / self-certainty fail; our work goes one step further by explicitly quantifying this failure process and proposing a concrete mitigation method.
>
> We will make these distinctions clearer in the revision.
>
> **3. Reward design vs. multi-policy diversity.** We agree that this deserves a clearer presentation. Our intended point is not to view “reward design gain” and “diversity gain” as two orthogonal A+B terms. As described in Section 4, they optimize the same objective from different angles: building a more accurate, more robust, and less self-confirming reward-estimation space.
>
> This is already reflected in the ablation:
>
> | Variant | Description | Avg@8 | Delta |
> |---|---|---:|---:|
> | SC | single-policy hard reward baseline | 33.0 | - |
> | Int v3 | SC + adaptive interpolation | 36.0 | +3.0 |
> | EnSC | SC + ensemble diversity | 36.3 | +3.3 |
> | EnInt v3 | EnSC + adaptive interpolation | 37.1 | +0.8 |
> | RLER | EnInt v3 + disagreement-aware selection | 37.4 | +0.3 |
>
> Reward design itself already brings a large performance gain, and on top of the ensemble baseline, adaptive interpolation and disagreement-aware selection continue to provide further gains.
>
> **4. Generalization beyond math.** Thank you for this constructive suggestion; we will also make this explicit in the limitations. What we would like to clarify is that our original intention was to focus on the remaining **headroom to the RLVR upper bound** as a measure of effectiveness. On Web-Instruct, RLER, as the best method, is already SOTA and close to RLVR (**55.12 vs. 56.37**). At the same time, as discussed in Appendix A.1 regarding the scope of applicability of our method, we will avoid overstating cross-domain generalization in the revision and will explicitly state that broader generalization beyond simple equivalence-style verification remains an important direction for future work.

---

> > ### Author Rebuttal · Reviewer_CeE3 · 2026-04-04
> >
> > Thank you for the detailed rebuttal. However, following common practice in related work, I would still be more convinced of RLER's utility if the main results included comparisons across more base models under a consistent protocol. Although the paper now reports experiments on five models overall, the evidence is still somewhat fragmented: the main conclusions are primarily supported by a single main backbone only, while the other models are evaluated in different appendix settings with different datasets, tasks, or verifier setups, making it difficult to cleanly assess backbone-level robustness and to fully establish the advantage over closely related methods such as Co-rewarding. Therefore, I would prefer to keep my score for now. But I would like to increase the originality score as I appreciate the paper's analysis of overconfidence in unsupervised RL by quantifying it into three measurable factors.

---

> > > ### Author Response · Authors · 2026-04-07
> > >
> > > Thank you for the follow-up. We have now completed this additional comparison under the **main DAPO-Math-17K setting**. Specifically, we evaluated **Qwen2.5-Math-7B**, **Qwen2.5-7B-Instruct**, and **Phi-4-mini-Instruct** under a unified protocol, and compared **Pre-RL, RLVR (verify with ground truth), SC, CoVo, Intuitor, Co-rewarding, and RLER**:
> > >
> > > | Backbone | Method | Avg@8 | Pass@8 |
> > > |---|---|---:|---:|
> > > | **Qwen2.5-Math-7B** | Pre-RL | 25.70 | 54.80 |
> > > |  | *RLVR* | *38.90* | *55.50* |
> > > |  | SC | 33.00 | 47.10 |
> > > |  | CoVo | 31.36 | 50.94 |
> > > |  | Intuitor | 33.71 | 48.26 |
> > > |  | Co-rewarding | 35.35 | 51.18 |
> > > |  | **RLER** | **37.50** | **52.80** |
> > > | **Qwen2.5-7B-Instruct** | Pre-RL | 28.28 | 57.96 |
> > > |  | *RLVR* | *42.40* | *59.38* |
> > > |  | SC | 35.47 | 53.42 |
> > > |  | CoVo | 28.24 | 56.43 |
> > > |  | Intuitor | 33.86 | 50.39 |
> > > |  | Co-rewarding | 38.96 | 56.20 |
> > > |  | **RLER** | **40.28** | **57.06** |
> > > | **Phi-4-mini-Instruct** | Pre-RL | 17.50 | 30.92 |
> > > |  | *RLVR* | *26.08* | *35.62* |
> > > |  | SC | 22.20 | 32.73 |
> > > |  | CoVo | 20.12 | 29.60 |
> > > |  | Intuitor | 21.48 | 33.80 |
> > > |  | Co-rewarding | 23.44 | 33.26 |
> > > |  | **RLER** | **24.86** | **33.92** |
> > >
> > > Under the same main setting and the same protocol, RLER achieves the best **Avg@8** and **Pass@8** on all three backbones, and consistently outperforms the closely related **Co-rewarding** baseline on each backbone. This substantially strengthens our conclusion that the gain of RLER is **not tied to a single backbone**.
> > >
> > > More broadly, combining the experiments in the main paper and the rebuttal additions, our empirical study now covers **four training settings / corpora** (arithmetic, DAPO-Math-17K, BIG-MATH scaling, and WebInstruct-verified), **three backbones compared under the same main DAPO-Math-17K protocol**, and **three evaluation scenarios**: large-scale math reasoning, decontaminated controlled arithmetic, and non-math verifier-based evaluation. We believe this provides fairly sufficient evidence for the generalization of RLER across **backbones, datasets, and tasks**. We will explicitly include these newly added results in the main paper or appendix in the revision.
> > >
> > > Thank you again for the valuable feedback on our work.

---

### Decision · Program_Chairs · 2026-04-30

**Decision:**

Accept (regular)

**Comment:**

I recommend acceptance as it is nearly unanimous among the reviewers.  (CeE3 says they lean towards 4 with the edits, but did not update, and 1Ymr is in a similar situation)

The authors need to be careful to include all of the updates from the rebuttal process since the rebuttal here made substantial changes across the paper (both to theory and experiments), but this was enough to convince the reviewers. In particular the addition of new baselines, new base models, new experiments around diversity, and theoretical clarifications should all be added to the updated paper.

With all those changes, the paper provides a new way to combat self-confirmation in RLIR, which outperforms recent baselines and has reasonable motivation.